# Rabies Vaccination and Public Health Insights in the Extended Arabian Gulf and Saudi Arabia: A Systematic Scoping Review

**DOI:** 10.3390/diseases13040124

**Published:** 2025-04-21

**Authors:** Helal F. Hetta, Khalid S. Albalawi, Amal M. Almalki, Nasser D. Albalawi, Abdulmajeed S. Albalawi, Suleiman M. Al-Atwi, Saleh E. Alatawi, Mousa J. Alharbi, MeshaL F. Albalawi, Ahmad A. Alharbi, Hassabelrasoul Elfadil, Abdullah S. Albalawi, Reem Sayad

**Affiliations:** 1Division of Microbiology, Immunology and Biotechnology, Department of Natural Products and Alternative Medicine, Faculty of Pharmacy, University of Tabuk, Tabuk 71491, Saudi Arabia; aam_alharbi@ut.edu.sa (A.A.A.); habdelgadir@ut.edu.sa (H.E.); 2Department of Supply Chain, Tabuk Health Cluster, Tabuk 47311, Saudi Arabia; khsaalbalawi@moh.gov.sa (K.S.A.); ammualmalki@moh.gov.sa (A.M.A.); pharmnasser23@gmail.com (N.D.A.); ag.majeed47@gmail.com (A.S.A.); sulemanma@moh.gov.sa (S.M.A.-A.); 3Primary Health Care, Tabuk Health Cluster, Tabuk 47311, Saudi Arabia; saaialatawi@moh.gov.sa; 4Ministry of Health Tabuk Region, Tabuk 47311, Saudi Arabia; moussa.jzh@gmail.com; 5Executive Management for Quality & Institutional Excellence, Tabuk Health Cluster, Tabuk 47311, Saudi Arabia; mefalbalawi@moh.gov.sa; 6Department of Pharmaceutical Chemistry, Faculty of Pharmacy, University of Tabuk, Tabuk 71491, Saudi Arabia; abs_albalawi@ut.edu.sa; 7Department of Histology, Faculty of Medicine, Assiut University, Assiut 71515, Egypt; reem.17289806@med.aun.edu.eg

**Keywords:** rabies, post-exposure prophylaxis, vaccination protocols

## Abstract

Background and Aim: This systematic scoping review examines rabies-related incidents, interventions, and post-exposure immunoprophylaxis in the Arabian Gulf region and Saudi Arabian Peninsula. Methods: A comprehensive literature search was conducted in databases including PubMed, Scopus, WoS, MedLine, and Cochrane Library up to July 2024. Studies were included discussing the reported cases of rabies that received the PEP in all countries of the Arabian Gulf, their epidemiological data, the received schedules of vaccination, and their safety. The search was done by using the following terminologies: rabies vaccine, rabies human diploid cell vaccine, vaccine, Saudi Arabia, Bahrain, Iraq, Kuwait, Oman, Qatar, United Arab Emirates, Southwest Asia, Iran, West Asia, Western Asia, Persian Gulf, Arabian Gulf, Gulf of Ajam, Saudi Arabian Peninsula, and The Kingdom of Saudi Arabia. Results: The systematic scoping review included 36 studies, synthesizing findings from diverse research designs, including large-scale cross-sectional studies and case reports, spanning nearly three decades. Findings indicated that young males in urban areas are most at risk for animal bites, predominantly from domestic dogs and cats. While post-exposure prophylaxis (PEP) was generally administered within recommended timeframes, vaccination completion rates varied. Conclusions: The review highlighted gaps in public awareness about rabies risks and prevention. Vaccine safety profiles were generally favorable, with mostly mild-to-moderate side effects reported. The study underscores the need for enhanced public health education, standardized PEP protocols, and a One Health approach to rabies prevention.

## 1. Introduction

An estimated 59,000 people die from rabies each year, a deadly illness brought on by the lyssaviruses and the rabies virus (RABV) [1]. Globally, there were 14,075 incident rabies cases and 13,743 deaths in 2019, both of which were lower than in 1990 [2].

Dog bites are the cause of up to 99% of human cases globally [3]. If left untreated, rabies can be lethal and is typically spread to humans and animals by bites, scratches, or contaminated mucus membrane from rabid animals [4,5,6]. Clinical rabies cannot be cured, but it can be easily avoided by giving prompt and sufficient post-exposure prophylaxis (PEP). PEP includes several rabies vaccinations, the delivery of rabies immunoglobulins (RIG), or, more recently, licensed monoclonal antibody products if necessary [7], as well as a thorough wound cleaning with water, detergent, and antiseptics [7]. The PEP approach varies according to the type of exposure, the patient’s immune status, and whether they have received a prior rabies vaccination [8]. A person who has previously had the rabies vaccination, either as part of a comprehensive pre-exposure prophylaxis course or as PEP, is referred to as a previously vaccinated person, per 2010 recommendations. RIG is not advised for individuals who have received rabies vaccinations in the past, even decades ago [7]. Booster shots are the only suggested course of action. They will stimulate the generation of antibodies and elicit an amnestic response.

Depending on the schedule, rabies vaccinations can be given intramuscularly (IM) or intradermally (ID). Since 1992, the World Health Organization (WHO) has recommended intradermal rabies immunization. When used, vaccination costs and doses can be lowered by 60–80%, particularly in high-throughput clinics [9]. The 2010 WHO-recommended vaccination schedules require up to five clinic visits spread over approximately one month to minimize unintended adverse effects [7]. The long duration of immunization can cause people who could be exposed to rabies frequently not to receive the entire course of vaccination, and also cause a change in the route of administration [10].

The largest nation on the Arabian Peninsula is Saudi Arabia, and little information has been released regarding the country’s rabies situation. According to earlier reports, foxes are the primary rabies reservoir and have been implicated in most animal bites that have affected humans, including dogs, cats, rodents, and humans [11,12]. According to recent reports, rabies poses a health concern to humans and animals nationwide and is also thought to be spread by wild canines [13]. In 2007, a study of 4124 camels in the Al Qassim region revealed a clinical rabies incidence of 0.2%, most likely due to wild dogs (70%) and foxes (17%) spreading the disease. In the Al-Qassim region, between 1997 and 2006, the diagnosis of rabies was verified in 26 dogs, 10 foxes, 8 camels, and 7 cats [13]. The Saudi Ministry of Environment, Water, and Agriculture (MEWA) and the Saudi Ministry of Health (MoH) have received reports of 11,069 animal bites on humans between 2007 and 2009. Dogs and cats accounted for 49.5% and 26.6% of all injuries, respectively. Mice and rats (12.6%), camels (3.2%), foxes (1.3%), monkeys (0.7%), and wolves (0.5%) were next in line for injuries, underscoring the significance of animal rabies surveillance and control programs as a critical component of the disease’s prevention [12]. Animal-related injuries continue to be a public health concern in Saudi Arabia, and all countries of the Arabic Gulf, where most human bites are caused by wild dogs, and most rabid animals are located there [12].

Therefore, obtaining more detailed information on the epidemiology of animal rabies in the Arabian Gulf will be essential. In this systematic scoping review, we aim to systematically scope and synthesize the existing literature on rabies PEP protocols, safety outcomes, and public health responses in the Arabian Gulf and Saudi Peninsula, to inform more effective regional strategies aligned with One Health principles.

## 2. Methods

### 2.1. Information Sources and Search Strategy

Considering the PRISMA (Preferred Reporting Items for Systematic Reviews and Meta-Analyses) extension for scoping reviews, a systematic scoping review of clinical trials was created [14]. More details of the research methodology were provided by the PRISMA checklist (Appendix A). We registered the protocol in PROSPERO, protocol number: CRD420251027233.

Databases from SCOPUS, PubMed, the Web of Science (WoS), Cochrane, and MedLine through WoS were examined up to July 2024. The terminologies rabies vaccine, rabies human diploid cell vaccine, vaccine, Saudi Arabia, Bahrain, Iraq, Kuwait, Oman, Qatar, United Arab Emirates, Southwest Asia, Iran, West Asia, Western Asia, Persian Gulf, Arabian Gulf, Gulf of Ajam, Saudi Arabian Peninsula, The Kingdom of Saudi Arabia, and comparative clinical studies were the terms used to review observational studies in all languages published up to July 2024. Then, we used the Boolean operators AND OR to search all databases. Details of the search strategy are mentioned in Appendix A.

### 2.2. Eligibility Criteria

Inclusion criteria: Patients who received post-exposure prophylaxis of rabies were included. All types of vaccines were included, such as purified chick embryo cell vaccine (PCECV), purified Vero cell rabies vaccine (PVRV), human diploid cell rabies vaccine (HDCV), and Vero rabies vaccine (PVRV). Prospective or retrospective trials were incorporated, such as randomized controlled trials, observational studies, case reports, case series, cross-sectional studies, or cohort studies. Moreover, we included letters to the editor that published results from clinical trials that met our inclusion criteria. The inclusion of the published studies was limited to the Arabic Gulf countries, Turkey, and Iran only. There was no restriction according to the language of the published trials.

Although Turkey and Iran are not geographically part of the Arabian Gulf or the Saudi Peninsula, their inclusion in this review is justified based on several key factors. Both Turkey and Iran are influential countries in the broader Middle East region, with historical, political, and economic ties to Gulf Cooperation Council (GCC) countries. Their health policies and practices often influence or align with regional trends. High levels of travel, trade, and workforce migration between Turkey, Iran, and Gulf countries facilitate shared public health challenges, including infectious disease control and rabies prevention strategies. Iran and Turkey face similar climatic, ecological, and zoonotic disease patterns, including rabies endemicity, particularly in rural or border areas. Their strategies for PEP, vaccination schedules, and health outcomes offer relevant insights for the Gulf region. Data from many Arabian Gulf countries may be limited due to underreporting or publication gaps. Including data from Iran and Turkey enhances the comprehensiveness of the review by providing comparative regional evidence on vaccine safety, efficacy, and scheduling. Both Iran and Turkey are among the most active countries in the Middle East in terms of biomedical research output. Their inclusion allows the review to incorporate valuable peer-reviewed literature and clinical data that may be applicable or adaptable to the Arabian Gulf context [15,16,17,18].

Exclusion criteria: Experimental studies, animal studies, and case reports of only one case. Additionally, we excluded studies that did not present any data about the status of vaccination for rabies victims.

### 2.3. Research Questions

This systematic scoping review aims to report the demographic data of the published studies, including rabies cases that received PEP, their vaccination schedule, and the safety of the included vaccines.

### 2.4. Trial Selection

After reading the abstracts and full texts, certain keywords prompted both researchers to choose the papers. The two researchers used the inclusion criteria to assess the trials. Subsequently, every abstract and full text were downloaded and evaluated independently based on the pre-established inclusion criteria. When there was disagreement among the researchers, the third author assessed the acceptability of the study.

### 2.5. Data Extraction

Two authors independently reviewed and evaluated each full text that met the inclusion criteria to be included in this systematic scoping review. Each investigator independently created a table that included the most crucial details from the chosen trials, and the outcomes were compared. They extracted the data in three tables: the first was the table of summaries of the included studies, the second was for baseline characteristics of the included patients, and the last was for the safety of the included vaccines.

### 2.6. Quality Assessment

Two authors independently evaluated the quality of each full text that met the inclusion criteria to be included in this systematic scoping review. The methodological index for non-randomized studies (MINORS) criteria were employed [19]. The maximum score was 24 for the comparative studies. Scores of 0–6 corresponded to very low quality, 7–10 corresponded to low quality, 11–15 corresponded to fair quality, and ≥16 corresponded to high quality.

### 2.7. Analysis of Outcome Measures

We conducted qualitative analysis by collecting and summarizing the available data from the included studies. Moreover, we summarized the quantitative data using descriptive analysis, providing better representative data. According to quantitative analysis, the contentious variables were represented by median and range, while the binary data were represented by frequency and percentage.

The data extracted from the included studies were analyzed narratively to identify trends and patterns related to rabies PEP and vaccine safety. Given the heterogeneity of study designs and outcomes, we performed a descriptive synthesis to provide an overview of the key findings across different populations, settings, and protocols.

We categorized studies based on key characteristics, such as age group, sex, type of area (urban vs. rural), and animal exposure type (e.g., domestic dogs, cats, and wild animals). These factors were analyzed to explore their relationship with PEP adherence and vaccine safety outcomes.

The timing of PEP initiation, specifically the duration between animal exposure and the administration of prophylactic treatment, was another key factor in the analysis. We examined the proportion of cases that received PEP within 48 h (the recommended timeframe) and compared this across urban and rural settings.

Variations in vaccination regimens, including the number of doses administered (3, 4, or 5 doses), were also examined. This was particularly important in understanding differences in clinical practice.

Data on adverse events related to the rabies vaccination were collated from studies that provided safety data. The analysis focused on the frequency and severity of reported side effects across various vaccine types (e.g., purified chick embryo cell vaccine and purified Vero cell vaccine).

We also analyzed the influence of geographic and ecological factors on rabies exposure. Data from Turkey and Iran, included due to their geographical proximity and similar epidemiological patterns, were integrated to understand the broader regional dynamics. These studies allowed us to explore rabies exposure risks in areas with significant cross-border movement, including the impact of migration and trade on rabies transmission dynamics.

By organizing the data in these thematic categories, we provided a comprehensive narrative synthesis of the epidemiological trends, vaccination practices, and vaccine safety outcomes across different healthcare settings. This analysis offers valuable insights into the effectiveness of PEP strategies and highlights key areas for improvement, particularly in rural settings and in regions with incomplete vaccination practices.

## 3. Results

### 3.1. Literature Search

A total of 992 articles from PubMed, Scopus, Web of Science, MedLine, and Cochrane Library were screened. After the removal of duplicates, a total of 619 articles were selected for title and abstract screening. After reviewing the titles and abstracts, 92 articles were selected for full-text review. From these studies, 56 were excluded from the review. Finally, 36 studies met our study’s inclusion criteria. The literature search is graphically represented in the PRISMA flow chart (Figure 1).

### 3.2. Study Characteristics

The results present a comprehensive collection of studies on rabies-related incidents and interventions, encompassing various research designs, including one prospective study [20], one randomized controlled trial [21], three case reports [22,23,24], three case series [25,26,27], eighteen cross-sectional studies [15,17,18,28,29,30,31,32,33,34,35,36,37,38,39,40,41,42], and eight retrospective analyses [16,43,44,45,46,47,48,49]. These studies were conducted across multiple countries, with a significant focus on Iran and Turkey, as well as contributions from Lebanon and Saudi Arabia. The research spans a wide timeframe, with studies dating from 1995 to 2024, covering periods ranging from a few months to several years. Collectively, these studies represent a substantial sample size of over 400,000 participants, with individual study samples varying from as few as 2 cases in some reports to as many as 260,470 in large-scale registry-based studies. More details are presented in Table 1.

### 3.3. Baseline Characteristics

In a comprehensive analysis of various studies on age demographics conducted between 1995 and 2024, it became evident that a wide range of age groups were studied, illustrating the diversity in the population samples. Most bite victims were young, with several studies reporting a higher incidence among males. In Khazaei et al.’s study (2023), the most affected age group was between 16 and 30 years [29], while Rasooli et al. (2020) found that all victims were between 10 and 67 years old [25]. Additionally, Khoubfekr et al. (2024) focused on ages 7 to 12, with a 100% male population [22]. In their study, Davarani et al. (2023) captured the distribution of age groups, prominently noting 65.3% for ages 7–12 years and a 66.5% male population [29]. On the other hand, Kassiri et al. (2018) examined diverse age ranges, with 76.6% male population, featuring categories from 0 to 4 years to above 61 years [35]. Notably, a 24.1% male population was reported by Oztoprak et al. (2021) in the 15-year-old and above age group [43].

Urban areas had a higher prevalence of bite incidents compared to rural areas, as seen in studies by Janatolmakan et al. (2020) and Amiri et al. (2020) [32,33]. This could be attributed to higher population densities and closer interactions with domestic animals in urban settings. These findings highlight the possibility that the remoteness of rural areas from health centers is a factor influencing the rate of receiving vaccinations. In other words, people in rural communities do not seek rabies treatment because of the distance and the perception that the risk of rabies is lower in rural communities.

The summarized data also provided insight into various studies on animal bite incidents, their implications for rabies vaccination, and demographic factors. Across these studies, domestic dogs were the primary animals involved in bite incidents, with cats being the second most common. Rodents, such as hamsters and mice, are considered low risk for rabies transmission due to their small size, susceptibility to fatal injuries during encounters with larger mammals, and lower likelihood of surviving rabies infection long enough to transmit the virus.

The studies revealed that the hands and upper limbs were frequently bitten, especially in cases involving dogs. For instance, Khoubfekr et al. (2024) found that dogs were responsible for all bites, primarily affecting the upper limbs [22], while other studies, such as those by Davarani et al. (2023) and Oztoprak et al. (2021) [28,43], showed that a significant proportion of bites occurred on the hands and lower limbs.

Lastly, in terms of injury type, the majority were punctures and scratches, with a few cases of more severe injuries like bone fractures. For example, Davarani et al. (2023) reported that puncture wounds accounted for 61.4% of injuries, while scratches were 36.9% [28]. The timing of post-exposure prophylaxis varied across studies, with most individuals receiving vaccination within 48 h. For instance, Khazaei et al. (2023) reported that 97.2% of cases received treatment within this timeframe, highlighting the urgency of rabies prevention after a bite [29]. More details are presented in Table 2 and Appendix A. A summary of the findings is presented in Table 3 and Figure 2.

### 3.4. Vaccination Schedules

The results presented a comprehensive overview of rabies PEP practices across various studies conducted primarily in the Persian Gulf countries, with a focus on Iran and Turkey, with participation from Saudi Arabia and Lebanon. The data consistently showed that the majority of patients received rabies vaccination within the first 48 h of exposure, with percentages ranging from 81.14% to 97.20% across different studies. The vaccination protocols typically involved either a 3-dose or 5-dose regimen, with some studies reporting the use of specific vaccines, such as Verorab, purified chick embryo cell vaccine (PCECV, Rabipur), and Vero rabies vaccine (PVRV).

The vaccination regimens varied, with a mix of complete and incomplete vaccination records. For example, Bay et al. (2021) noted that only 12.8% of bite victims received the full four-dose vaccine [15], while a significant portion did not require vaccination. This variability underscores the importance of timely and appropriate post-exposure prophylaxis based on the type and severity of the bite.

In addition to vaccination, many studies report on the administration of rabies immunoglobulin (RIG), with rates varying significantly between studies, ranging from 29.06% to 78.1% of cases. For instance, Rahmanian et al. (2020) reported RIG administration in 29.06% of cases [45], while Can et al. (2020) noted a higher rate of 78.1% [44]. The timing of RIG administration was also highlighted, with Davarani et al. (2023) reporting that 68.9% of patients received RIG within the first 12 h post-exposure [28]. Several studies also mentioned tetanus vaccination and antibiotic prophylaxis as part of the treatment protocol. Notably, some studies, such as Amiri et al. (2015) [37], explored patients’ awareness and reasons for seeking treatment, revealing that only about half of the patients were aware of the need for rabies vaccination after a dog bite, which means gaps in public knowledge about rabies PEP. Overall, the data reflected a general adherence to the WHO guidelines for rabies PEP, with variations in practice across different healthcare settings and regions. More details are presented at Table 2 and Appendix A.

### 3.5. Safety of the Included Vaccines

Seri et al. (2014) [20] compared the safety of two vaccines, Verorab and Abhayrab. Regarding Verorab, the most common side effects were fever (4.99%), weakness (4.99%), headache (3.55%), and local pain (3.02%). On the other hand, the less common side effects were nausea, abdominal pain, and various other mild symptoms. There were no reports of swelling, bruising, insomnia, numbness, irregular menstruation, or decreased libido. Regarding Abhayrab, it generally had a higher incidence of side effects compared to Verorab. The most common side effects were headache (24.78%), fever (15.15%), local swelling (10.28%), and local pain (9.4%). There were higher rates of dizziness, itching, nausea, and other symptoms compared to Verorab.

Ramezankhani et al. (2016) [21] compared PVRV and PCECV.

For PVRV, the most common side effects were local pain (3.9%), fever (1.9%), and headache (1.4%). There were no reports of bruising, itching, or hypotension. On the other hand, the most common side effects of PCECV were local pain (3.8%), bruising (2.5%), and weakness (1.7%), a generally similar side effect profile to PVRV, with some variations (Table 4).

### 3.6. Quality Assessment

The methodological quality assessment using the MINORS criteria revealed a variable but generally acceptable risk of bias across the 36 included studies. Overall, scores of non-comparative studies ranged from 12 to 15 out of a possible 16 points. The majority of studies scored between 13 and 15 points, suggesting a high quality with low risk of bias.

Overall, scores of the 3 comparative studies ranged from 23 to 24 out of a possible 24 points. The three studies demonstrated high methodological quality, indicating low risk of bias. More details of the assessment of the included studies are presented in Table 5.

## 4. Discussion

### 4.1. Demographics of the Included Patients

This comprehensive review of rabies-related incidents and interventions in the Extended Persian Gulf region, particularly in Iran and Turkey, with contributions from Lebanon and Saudi Arabia, provided valuable insights into the epidemiology of animal bites and the implementation of PEP protocols. The diversity of study designs and settings suggests a comprehensive approach to understanding rabies-related issues in the Extended Persian Gulf countries. The prevalence of cross-sectional and retrospective studies indicates a focus on epidemiological patterns and trends in rabies exposure and treatment. Notable aspects include the presence of several large-scale studies, such as the registry-based cross-sectional study by Khazaei et al. (2023) with 260,470 participants, which likely provides valuable insights into the broader patterns of rabies incidents [29]. Additionally, the inclusion of case reports, particularly those focusing on rare or unique presentations (e.g., Khoubfekr et al., 2024; Ansari et al., 2011), highlights the importance of documenting unusual cases for medical education and awareness [22,23]. The consistent research efforts over nearly three decades demonstrate an ongoing commitment to understanding and addressing rabies-related health concerns in the region.

The studies consistently showed that young males aged 20–39 years are disproportionately affected by animal bites. This trend could be attributed to higher risk-taking behaviors or increased outdoor activities among this demographic, where encounters with animals are more frequent. Meanwhile, children, particularly those aged 0–12 years, showed significant exposure, especially in regions with prevalent stray dog populations, as seen in studies like that of Davarani et al. (2023) [28], who reported 65.3% exposure in 7–12-year-olds. On the other hand, the elderly (60+ years) exhibited lower but still non-negligible exposure, possibly due to reduced mobility or residence in rural areas where animal interactions are more common. These age-related trends highlight the dynamic interplay between lifestyle, geography, and the risk of rabies, stressing the need for targeted interventions across different population groups. The prevalence of bites to the hands and upper limbs indicates a need for public education on proper behavior around animals to minimize such incidents.

The prevalence of bites in urban areas suggests that urbanization may be a factor in human–animal interactions, possibly due to higher population densities and closer proximity to domestic animals. Domestic dogs emerged as the primary culprits in bite incidents, followed by cats. This finding highlights the importance of responsible pet ownership and vaccination programs for domestic animals.

Cases of aggression by rodents, such as hamsters and mice, warrant special consideration in rabies risk assessment. Rodents, particularly small ones, are rarely associated with rabies transmission to humans. This is partly because they typically succumb to the injury from rabid animal attacks, limiting their potential as carriers of the virus. Their ecological role and behavior also contribute to the low transmission risk—rodents are usually prey animals with limited interaction with typical rabies reservoirs, like canines, bats, and other carnivores known to transmit the virus. In discussing cases from studies like those of Rahmanian et al. (2020), Davarani et al. (2023), and Porsuk et al. (2021), public health guidelines often recommend that rodent bites do not typically require PEP unless there are unusual circumstances, such as the animal displaying abnormal behavior or originating from an environment with known rabies outbreaks [22,28,31].

Highlighting these cases also provided an opportunity to discuss regional variations in PEP guidelines. Some regions may advocate a more cautious approach, while others rely on data indicating that rabies transmission from rodents to humans is exceedingly rare. Additionally, these cases allowed us to examine whether any deviations from standard PEP protocols were applied in response to rodent bites and whether additional clinical or observational criteria—such as the aggressor’s health status or species-specific behavior—were considered.

### 4.2. Protocols of Vaccination

The majority of bite victims received PEP within 48 h of exposure, which aligns with the WHO recommendations for timely intervention. However, the variability in vaccination completion rates highlights a potential area for improvement in patient follow-up and education about the importance of completing the full course of vaccination.

The studies revealed a mix of 3-dose and 5-dose vaccination regimens, reflecting evolving guidelines and potentially differing resources across healthcare settings. The use of specific vaccines, like Verorab, PCECV, and PVRV, indicated adherence to internationally recognized standards for rabies prevention. The wide range in RIG administration rates (29.06% to 78.1%) suggested significant variability in clinical practice or resource availability across different healthcare settings. This disparity warrants further investigation to ensure equitable access to optimal care.

The review highlighted significant variations in clinical practices, especially regarding incomplete immunization in rabies PEP. Studies show that a substantial proportion of patients received incomplete immunization. For instance, Kassiri et al. (2018) reported that approximately 38.6% of patients in their study received a complete vaccine regimen, with the majority (61.4%) receiving incomplete doses [35]. Similarly, a large proportion of cases in Porsuk et al. (2021) and Can et al.’s (2020) studies involved incomplete vaccination protocols [31,44]. Incomplete immunization is a critical concern, as it compromises the effectiveness of rabies prevention, increasing the risk of rabies transmission, particularly in regions with high animal exposure. Addressing this issue in clinical practice requires standardized PEP protocols, improved patient education, and enhanced access to vaccination to ensure timely and complete immunization.

The discrepancies in vaccination completion and RIG use can be attributed to several key factors, including limited access to healthcare services in rural settings, which leads to delays in initiating PEP. Studies like those of Sarbazi et al. (2020) and Poorolajal et al. (2015) showed that rural regions experience significant delays in receiving timely PEP, which can result in incomplete vaccination [40]. Inconsistent application of vaccination protocols contributes to incomplete vaccination regimens. This discrepancy is likely due to differing practices and resources available across regions. In some instances, patients may not complete the full vaccine regimen due to logistical issues or a misunderstanding of the importance of completing the full series.

The need for RIG is a critical component of PEP when the exposure is considered high risk, such as when the bite is from a wild animal or involves multiple puncture sites. Rahmanian et al. (2020) and Davarani et al. (2023) both highlighted how RIG use varies depending on the severity of the injury and the type of animal involved [22,28]. However, the failure to administer RIG in some cases may be linked to limited knowledge or a lack of resources in certain healthcare settings.

Non-adherence to treatment schedules can also be attributed to patient-related factors, such as fear of side effects, misunderstanding the importance of completing the vaccine regimen, or logistical challenges in returning for multiple doses. This was particularly noted in studies such as that of Yıldırım et al. (2022), who reported that nearly 36.2% of patients did not complete their vaccinations [30].

In some studies, the underreporting of animal bites or delays in seeking medical care may also skew data on the completion of PEP and RIG use. Rural settings often have fewer health infrastructure resources, leading to missed opportunities for PEP administration or RIG use, contributing to incomplete vaccination courses.

The fact that only about half of the patients were aware of the need for rabies vaccination after a dog bite reveals a critical gap in public awareness. This underscores the need for enhanced public health education campaigns to improve awareness of rabies risks and the importance of seeking prompt medical attention after animal bites.

### 4.3. Different Practices of Post-Exposure Prophylaxis (PEP)

PEP initiation times and vaccination regimens differed significantly between urban and rural settings. Urban centers, with better healthcare infrastructure and access to vaccines, tended to initiate PEP more promptly. In contrast, rural areas may face delays in initiating treatment due to healthcare access issues or delayed reporting of animal bites. For instance, Davarani et al. (2023) found that 93% of patients in urban areas received treatment within 48 h, compared to lower rates in rural regions [28]. This delay increases the risk of incomplete immunization, which is more common in rural settings due to limited healthcare resources.

The type and number of doses administered varied across different healthcare settings. While Oztoprak et al. (2021) reported a higher frequency of full 5-dose vaccination schedules (40.5%) in urban settings [43], Amiri et al. (2020) found that a significant portion of rural patients received incomplete vaccinations [37]. This reflects the variation in adherence to international vaccination guidelines and the challenges in ensuring full immunization, particularly in areas with fewer resources.

The type of animal involved in the exposure also influenced PEP protocols. In countries like Turkey and Iran, where rabies is endemic in stray dog populations, the majority of exposures are related to dogs (67.7% in Davarani et al.’s study and 61.2% in Oztoprak et al.) [28,43], which are treated according to standard PEP protocols. However, exposures from wild animals, like bats or foxes, may lead to more aggressive treatments due to their higher risk of rabies transmission. This differentiation is crucial in determining the type and urgency of the prophylactic regimen.

The timeliness of PEP is another significant factor that varies by setting. For instance, Sarbazi et al. (2020) and Poorolajal et al. (2015) report that the majority of patients in urban settings received PEP within 48 h, a key window for effective rabies prevention [34,40]. However, delays of over 48 h, particularly in rural regions, are common due to difficulties in timely reporting and accessing medical treatment.

These distinctions highlight the need for improved healthcare access, particularly in underserved areas, and the importance of standardized PEP protocols across different settings to ensure timely and effective rabies prevention.

### 4.4. WHO’s Global Rabies Elimination Strategy (2018)

The findings of the review are aligned with and further inform the WHO’s Global Rabies Elimination Strategy and its guidelines on rabies PEP [3]. The WHO recommends that PEP be initiated as soon as possible, ideally within 24 h of exposure, and no later than 48 h [3]. Our review found that urban areas generally adhered to these timelines, with over 90% of patients in studies, such as those of Davarani et al. (2023) and Khazaei et al. (2023), receiving PEP within 48 h [28,29]. However, in rural settings, delays in initiation were noted, reflecting challenges in healthcare access and infrastructure. These findings emphasize the need for strengthening rabies surveillance and improving access to timely medical interventions, as emphasized by the WHO’s call for better global rabies surveillance systems.

According to the WHO guidelines, completing a full five-dose vaccine regimen is essential for rabies prevention [3]. In our review, incomplete vaccination was a recurring issue, particularly in rural regions, with studies such as those of Amiri et al. (2020) and Porsuk et al. (2021) showing that a significant proportion of patients received fewer than five doses [31,37]. This is consistent with the WHO’s recommendation to monitor vaccine adherence and address barriers to completing vaccination schedules. Improving community education and ensuring supply chain reliability for vaccines are key areas to improve completion rates.

The WHO advocates for equitable access to vaccines and PEP across all regions, especially in rural and underserved areas [3]. Our findings underscore the need for increased efforts in these areas, where delays in treatment and incomplete immunization were more common. Programs to improve access to rabies vaccines in low-resource settings are essential to reduce the rabies burden, which aligns with the WHO’s emphasis on targeted strategies for rural and high-risk populations.

Lastly, the WHO’s One Health approach to rabies prevention integrates human, animal, and environmental health. Our review’s inclusion of Turkey and Iran, along with Gulf countries, highlights the transnational nature of rabies risks and the importance of regional cooperation. Data from these countries can help inform cross-border rabies control strategies and improve vaccine distribution and education. This is consistent with the WHO’s strategy to enhance collaborative efforts in rabies prevention across countries with shared risks and health challenges.

### 4.5. Safety of Vaccines

The overall safety of reported vaccines appears to be generally safe, with mostly mild-to-moderate side effects reported. Verorab appeared to have a better safety profile than Abhayrab, with lower incidence rates for most side effects. PVRV and PCECV showed similar safety profiles, with some variations in specific side effects [20,21]. Local pain at the injection site was consistently reported across all vaccines. Systemic effects, like headache, fever, and weakness, were also commonly reported. Serious side effects appeared to be rare, with no life-threatening reactions reported in these studies. Some effects, like irregular menstruation and decreased libido, were only reported in the Abhayrab group, but at very low rates. The two studies showed some differences in reported side effects, which could be due to differences in study design, population, or reporting methods. The studies had different sample sizes and may have used different methods for collecting and reporting side effects. While all vaccines showed some side effects, they appeared to be generally safe and well tolerated. In the second study, Verorab, PVRV and PCECV demonstrated favorable safety profiles. However, individual responses to vaccines can vary, and patients should be informed about potential side effects. The choice of vaccine may depend on various factors, including availability, cost, and individual patient characteristics.

The WHO’s guidelines highlight the safety of rabies vaccines, reporting that adverse events are generally rare and mild. In our review, most studies, including those of Yıldırım et al. (2022) and Davarani et al. (2023), reported that the adverse events were mostly local reactions, such as pain at the injection site or mild fever [28,30], which is consistent with the WHO’s safety profiles for both purified chick embryo cell and purified Vero cell vaccines [3]. This reinforces the ongoing commitment to using safe vaccines as part of rabies control programs globally.

## 5. Limitations

The heterogeneity in study designs, sample sizes, and timeframes presents challenges in drawing definitive conclusions. Future research should focus on standardizing data collection methods and conducting longitudinal studies to better understand trends over time. Additionally, investigating the reasons behind incomplete vaccination courses and variations in RIG administration could inform targeted interventions to improve PEP adherence. While the findings demonstrated general adherence to the WHO guidelines for rabies PEP, they also revealed areas for improvement in public education, healthcare practices, and resource allocation. Addressing these gaps could significantly enhance rabies prevention efforts in the Persian Gulf region and beyond.

## 6. Implications and Future Directions

There is a critical need for enhanced public health education campaigns to improve awareness of rabies risks and the importance of seeking prompt medical attention after animal bites. Targeted education efforts should focus on young males, who are disproportionately affected by animal bites. The higher prevalence of bites in urban areas suggests a need for improved urban planning and animal control measures to manage human–animal interactions in densely populated areas. There is also a need for strengthened domestic animal vaccination programs, particularly for dogs and cats, to reduce the risk of rabies transmission. Efforts should be made to standardize PEP protocols across healthcare settings, particularly regarding the administration of rabies RIG. Improved patient follow-up systems are needed to ensure completion of full vaccination courses. The variability in RIG administration rates suggests a need for a more equitable distribution of resources across healthcare settings to ensure all patients receive optimal care.

Future research should focus on standardizing data collection methods and conducting longitudinal studies to better understand trends over time. Investigation into the reasons behind incomplete vaccination courses could inform targeted interventions to improve PEP adherence. Findings could inform the development of more targeted and effective rabies prevention policies at local and national levels.

The findings underscore the importance of a One Health approach, integrating human health, animal health, and environmental factors in rabies prevention strategies. Given the regional focus of the studies, there is an opportunity for increased international collaboration in rabies prevention efforts across the Persian Gulf countries. These implications highlight the multifaceted approach needed to improve rabies prevention and control, encompassing public education, healthcare practices, policy development, and research priorities.

## Figures and Tables

**Figure 1 diseases-13-00124-f001:**
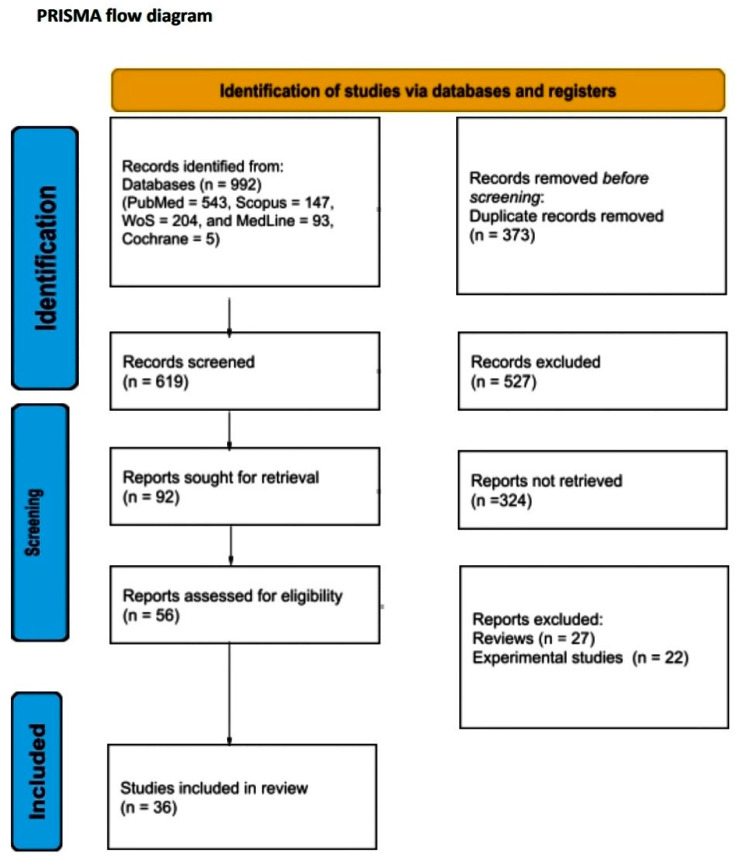
PRISMA flow diagram.

**Figure 2 diseases-13-00124-f002:**
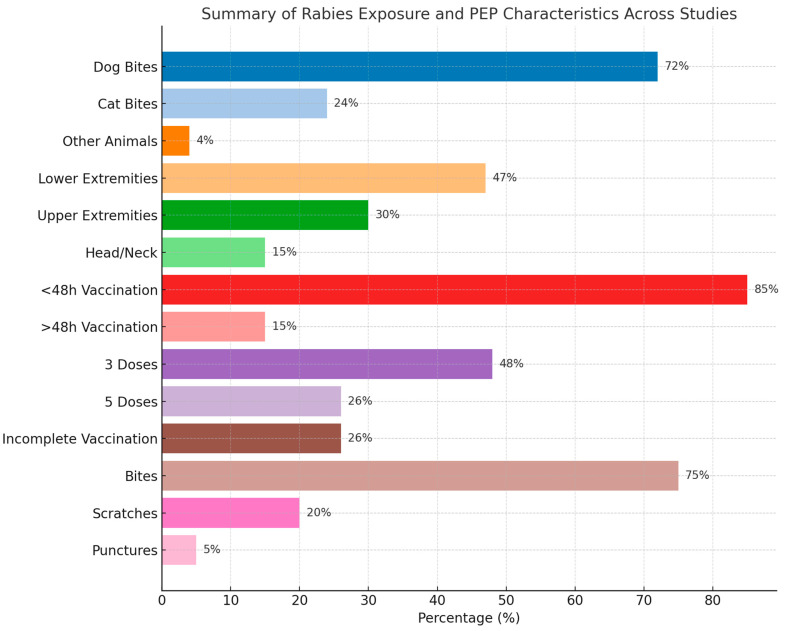
Summary of the baseline characteristics of the included patients.

**Table 1 diseases-13-00124-t001:** Summaries of the included studies. Abbreviations—ERIG: equine rabies immunoglobulin, PCECV: purified chick embryo cell vaccine, PVRV: purified Vero cell rabies vaccine, HDCV: human diploid cell rabies vaccine, RVNA: rabies neutralizing antibodies, PVRV: Vero rabies vaccine, and RVCs: rabies vaccination centers.

Study ID	Study Design	Setting (Country)	Date (Period of the Study)	Sample Size	PEP: Vaccine (Dose/Route of Administration/Number of Cycles)/Immunoglobulin. N (%)	
Khoubfekr et al., 2024 [22]	Two case reports	Iran	NA	2	1st case: The boy received RIG (CSL Behring 300 IU/2 mL, Marburg, Germany, Serums and Vaccines Ltd., Ambernath) at a dose of 20 IU/kg, totaling approximately 100 IU. Additionally, Verorab 0.5 mL was administered IM in the deltoid region. On day 3, he received the second dose of the vaccine, followed by the third dose on day 7. Additionally, he received a 4th dose of the vaccine on the 14th day post-exposure. 2nd case: Swift action was taken, and within 30 min of washing the affected areas, rabies immunoglobulin (CSL Behring 300 IU/2 mL, Marburg, Germany Serums and Vaccines Ltd., Ambernath) at a dose of 20 IU/kg, totaling approximately 100 IU, was administered. A rabies vaccine, Verorab 0.5 mL, was also administered IM in the deltoid region.	
Davarani et al., 2023 [28]	A cross-sectional descriptive-analytical study	Rabies prevention and treatment centers in Kerman, Iran	From April 2019 to March 2021	933	In patients who received the immunoglobulin injection before the end of the first 12 h: 381 (68.9) cases. In patients who received the immunoglobulin injection within 12–72 h: 151 (27.3) cases. Patients who received the immunoglobulin injection within 4–7 days 21 (3.8) cases. Patients who received rabies vaccine before the end of the first 48 h 868 (93) cases. Patients who received rabies vaccine within 48–72 h: 52 (5.6) cases. Patients who received rabies vaccine within 4–10 days: 12 (1.3) cases. Patients who received rabies vaccine within 11–20 days: 1 (0.1) case.	
Khazaei et al., 2023 [18]	Registry-based cross-sectional study	The rabies treatment centers located in the health centers of Iran	From March 2021 to March 2022	260,470	Patients who received vaccinations before the end of the first 48 h: 253,185 (97.20) cases. Patients who received vaccination after the first 48 h: 7285 (2.80) cases.	
Yıldırım et al., 2022 [30]	A retrospective cross-sectional study	Medical Faculty Rabies Vaccine Center, Ordu University, Ordu, Turkey	From 2014 to 2018	748	Patients who received the rabies vaccine: 477 (63.8) cases. Patients who did not complete or did not receive tetanus immunization: 2 (0.3) cases. Patients whose tetanus immunization was not known: 742 (99.2) cases. Patients who had complete tetanus immunization: 4 (0.5) cases.	
Bay et al., 2021 [15]	A cross-sectional study	Golestan Province, Iran	From March 2019 and March 2020	12,181	Patients who needed vaccines: 10,675 (99.7) cases. Patients who received a complete vaccine series: 1591 (13.1) cases. Patients who received the full vaccine four times: 10,608 (87.2) cases. Patients who received HRIG: 9840 (81.1) cases. Patients who received the DT vaccine: 6831 (56.1) cases.	
Celiloglu et al., 2021 [38]	Cross-sectional study	Adana City Training and Research Hospital, Turkey	From September 2017 to September 2018	2068	In terms of serial prophylaxis vaccinations, 761 (83.99) families were in full compliance, and 145 (16) families had discontinued vaccinations. RIG was administered to 447 (49.3) and not administered to 459 (50.7).	
Oztoprak et al., 2021 [43]	A retrospective descriptive study	Health Science University, Antalya Training and Research Hospital, Department of Infectious Diseases and Clinical Microbiology, Antalya, Turkey	From 2010 to 2013	2513	Vaccination was needed in 2015 cases (80.2). Rabies vaccine was administered to 1017 (40.5) cases as 5-dose. Rabies vaccine was administered to 626 (24.9) cases as 4-dose. Rabies vaccine was administered to 372 (14.8) cases as 2-1-1 schema.	
Porsuk et al., 2021 [31]	A descriptive cross-sectional study	Hospital’s emergency service in Turkey	From 2015 to 2019	3378	Post-exposure prophylaxis was applied to nearly all cases, 3352 (99.2), but 26 (0.8) of them were found to be inappropriate according to guidelines. HRIG treatment was applied together with the vaccine. One dose of vaccine to 256 (7.6) cases. Two doses were applied to 215 (6.4) cases. Three doses were applied to 1129 (33.4) cases. Four doses were applied to 653 (19.3) cases. Five doses were applied to 1125 (33.3) cases.	
Amiri et al., 2020 [32]	A cross-sectional study	Health centers of Najafabad in Isfahan province, Iran	From 2012 to 2017	4104	Patients who received the anti-rabies vaccine three times: 2197 (53) cases. Patients who received the anti-rabies vaccine five times: 1597 (38) cases. In terms of serum therapy, 3951 (96) of the injured did not need to receive rabies serum.	
Can et al., 2020 [44]	A retrospective cohort Study	Erzurum Palandöken State Hospital Emergency Service located in northeast Turkey	From August 2013 to June 2017	691	Human diploid cell vaccine. Single dose occurred for 114 (16.5) cases. There were two doses for 71 (10.3) cases. Three doses were administered to 150 (21.7) cases. 128 (18.5) cases received 4 doses. In 227 (32.9) cases, 5 doses were administered. In addition, human rabies immunoglobulin was applied to 540 (78.1) cases in the prophylaxis program.	
Janatolma Kan et al., 2020 [33]	A cross-sectional study	Kermanshah Province, Iran	From 2013 to 2017	5618	Most victims (n = 4594 (82)) had been vaccinated with the rabies vaccine three times. The remaining victims (n = 1024 (18)) had been vaccinated with the rabies vaccine five times.	
Rahmanian et al., 2020 [22]	A Retrospective Observational Study	Larestan County in the south Fars province, Iran	From 21 March 2018 to 20 March 2019	375	Patients who received three doses of vaccine: 268 (71.4) cases. Patients who received 5 doses of vaccine: 107 (28.5) cases. Patients who received 1 dose of RIG: 109 (29.06) cases.	
Rasooli et al., 2020 [25]	Case Series	Different regions of Iran	From 2014 to 2018	7	PVRV is an inactivated vaccine derived from the rabies virus cultured in PVRV cells. Combining this rabies vaccine and RIG as the recommended PEP protocol based on WHO guidelines to prevent rabies after exposure.	
Sarbazi et al., 2020 [34]	A cross-sectional study	Rabies center of Tabriz, Tabriz, Iran	From 1 March 2013 to 29 February 2019	3032	Patients who received vaccinations before the end of the first 48 h: 2773 (92.5). Patients who received vaccinations after the first 48 h: 259 (8.5).	
Hamta et al., 2019 [48]	A descriptive study	Centers for Disease Control and Prevention unit of Qom Provincial Health Center, Iran	From January 2017 to December 2018	2414	A delay of more than 48 h in the initiation of PEP was estimated among 305 (12.73) animal bite victims. Patients who received vaccinations before the end of the first 48 h: 2109 (87.36) victims.	
Kassiri et al., 2018 [35]	Cross-sectional study	Disease Prevention and Control Department of East Ahvaz Health Center, Southwestern Iran	From 2011 to 2013	2493	Treatment in 61.4% of victims suspected of having rabies who visited rabies treatment and prevention centers was done with three doses of anti-rabies vaccine and stopped after ten days. Treatment with five doses of anti-rabies vaccine was conducted in 38.6% of cases per year.	
Khazaei et al., 2018 [29]	A cross-sectional study	Nahavand district, Iran	From March 2015 to March 2017	1448	Patients who received vaccinations before the end of the first 48 h: 1175 (81.14) cases. Patients who received vaccination after the end of the first 48 h: 273 (18.85) cases.	
BabazadeH. et al., 2016 [36]	A cross-sectional study	Chalderan City, West Azerbaijan province, Iran	From 21 March 2008 to 20 March 2014	1449 (747 vs. 702).	Patients who received 3 doses of vaccine: 1607 (93.2). Patients who received 5 doses of vaccine: 117 (6.8).	
Ramezankhani et al., 2016 [21]	A double-blind randomized clinical trial	9 cities of Iran	2010	1449 (747 vs. 702)	PCECV (Rabipur^®^, Novartis, Germany)	PVRV (Verorab^®^, Mérieux Institute, France)
Amiri et al., 2015 [37]	A cross-sectional study	Guilan Province, north of Iran	From June 2011 to May 2012	1771	Only 810 cases (49.3%) were aware that they should receive the rabies vaccine after a dog bite. About 477 cases (29) were referred to RVCs after attending rural or urban health centers for receiving the tetanus vaccine or wound treatments. Then, 171 cases (10.4%) and 71 cases (4.7%) were referred to RVCs after attending private clinics and hospitals for receiving wound treatment, respectively. About 102 cases (6.2%) attended RVCs because their family or friends recommended them, and 6 cases (0.4%) attended RVCs after receiving advice from pharmacies, schools, or veterinarian clinic staff.	
Riabi et al., 2015 [39]	Cross-sectional study	Gonabad, Iran	From 2011 to 2013	616	517 (83.9) victims had been incompletely vaccinated, and 99 (16.1) received a complete vaccination.	
Poorolajal et al., 2015 [40]	Six-year population-based cross-sectional study	The Research Committee of Hamadan University of Medical Sciences, Tehran Province, Iran	From April 2006 to March 2012	22,766	Patients affected by animal bites and referred to health centers within 48 h received intramuscular doses of 0.5 mL. Those who were referred to health centers after 48 h received an IM dose of 1 mL, given in 5 doses over 4 weeks. Doses should be received on days 0, 3, 7, 14, and 30. Single or multiple dermal bites or wounds in the head and neck require immediate vaccination and RIG administration. Human RIG is given in a single dose of 20 IU per kg of body weight.	
Farahtaj et al., 2014 [26]	Case series	Multiple villages in Iran	From 2002 to 2011	16	Patients who received vaccination as a single shot: 1 (6.25) case. Patients who received vaccination as three shots: 2 (12.50) cases. Patients who received four vaccination shots: 13 (81.25) cases.	
Karbeyaz et al., 2014 [50]	A descriptive study	Director of Forensic Medicine, Eskisehir, Western Turkey	From 1 January 2006 to 31 December 2010	328	Patients who received vaccination against rabies: 42 (12.8) cases. 38 (11.6) cases of tetanus vaccination have been reported. Patients who did not receive vaccination: 16 (4.9) cases. There were 270 (82.3) cases of unknown vaccination status.	
Seri et al., 2014 [20]	A prospective trial.	The Infectious Diseases and Clinical Microbiology Outpatient Clinic of the Ministry of Health Ankara Training and Research Hospital, Turkey	From February 2010 to December 2010	1685 (761 vs. 924).	Abhayrab vaccine (Human Biologicals Institute, Ooty, India), each dose of vaccine as 0.5 mL into the deltoid muscle (924 patients). Vaccination at days 0, 3, 7, 14, and 28, plus immunoglobulin at initiation by scheduling 5 doses of vaccine and ERIG (Equirab; Bharat Serums and Vaccines Ltd., Navi Mumbai, India). OR Vaccination at days 0, 7, and 21 by a schedule of 2-1-1.	Verorab vaccine (Pasteur Merieux, Lyon, France), each dose of vaccine as 0.5 mL into the deltoid muscle (761 patients). Vaccination at days 0, 3, 7, 14, and 28, plus immunoglobulin at initiation by scheduling 5 doses of vaccine and ERIG (Equirab; Bharat Serums and Vaccines Ltd., India). OR Vaccination at days 0, 7, and 21 by a schedule of 2-1-1.
Charkazi et al., 2013 [41]	A descriptive cross-sectional study	Rabies Center of AqQala city, Northen of Iran	From 1998 to 2009	13,142	Patients who received a complete vaccination: 6463 (72) cases. Patients who receive incomplete vaccinations: 6679 (28) cases.	
Taghvaii et al., 2013 [17]	A cross-sectional study	Three health centers in Mashhad, Iran	From 2006 to 2009	14,037	Patients who received incomplete vaccinations: 11,672 (83.1) cases. Patients who received a complete vaccination: 2365 (16.9) cases.	
Ghannad et al., 2012 [42]	A cross-sectional descriptive study	Health centers located in Ilam Province, Iran	From April 1999 to March 2008	4420	There were 3596 (81.3) cases of incomplete vaccinations. Patients who received complete vaccination comprised 824 (18.7) cases.	
Ansari et al., 2011 [23]	Case reports	Sports Medicine Research Center, Tehran University of Medical Sciences, Tehran, Iran	NA	2	PVRV and HRIG, 20 IU/kg. Prophylaxis with doxycycline (100 mg, twice daily) and clindamycin (450 mg, three times daily) were administered for 7 days. The 5-dose regimen of the PEP rabies vaccine was completed (on days 0, 3, 7, 14, and 28).	PVRV and HRIG (20 IU/kg) were administered. The patient received antibiotic prophylaxis with oral amoxicillin-clavulanate (675/125 mg, twice daily). The 5-dose regimen of the PEP rabies vaccine was completed.
Bijari et al., 2011 [46]	A retrospective study	Health Center of Birjand University of Medical Sciences, Ghafari St., Birjand, Iran	From April 2002 to April 2009	1662	1597 (96.1) of cases were vaccinated within the first 24 h. 1361 (81.9) had incomplete vaccination. 301 (18.1) had complete vaccination.	
Najafi et al., 2009 [49]	Descriptive study	Mazandaran Province, northern Iran	From 2001 to 2005	32,079	Either immune globulin prophylaxis or a vaccine was used for PEP. Both HRIG and vaccines were given in 5263 cases (16.4%). Only vaccine was given in 26,816 cases (83.6).	
Sheikholeslami et al., 2009 [51]	Cross-sectional study	Rafsanjan, southeast of the Islamic Republic of Iran	From 2003 to 2005	1542	Rabies vaccine was given to 1311 cases (85). Rabies vaccine plus rabies immunoglobulin was given to 23 cases (15). The tetanus toxoid vaccine was given to 1018 cases (66).	
Kilic et al., 2006 [16]	A descriptive study	Narlidere District, Turkey	From 1999 to 2001	1569	1067 cases (68) were included in a post-exposure rabies vaccination program.	
Sengoz et al., 2006 [47]	A retrospective study	Hospital’s Center for Rabies Vaccination, Haseki Hospital for Training and Research, Istanbul, Turkey	From January 2003 to December 2003	7266	Patients who received vaccines with the first day: 5320 (72) cases. Patients who received vaccines with the 1st–5th day: 1890 (26) cases. Patients who received vaccines after the 5th day: 146 (2) cases. Patients who received 3 doses in 10 days: 2690 (37) cases. Patients who received 5 doses in 30 days: 1050 (14) cases. Patients who received the 2-1-1 schedule: 1770 (24). Number of patients not completing the standard vaccination schedule: 1750 (24) cases. Number of patients who did not require vaccine: 6 (<1) cases.	
Bizri et al., 2000 [27]	Case series	The American University of Beirut Medical Center (AUBMC), in Lebanon	From 1991 to 1996	8	Patients who received vaccination: 1 (12.5) case. There were 7 (87.5) cases of patients who did not receive vaccination.	
Tabbara et al., 1995 [24]	Two case reports	Department of Ophthalmology, College of Medicine, King Saud University	NA	2	The patient was given tetanus prophylaxis and 20 IU/kg of body weight of HRIG, one-half around the bite wound and the other half IM. She was also given HDCV and prophylactic antibiotics in the form of oral ampicillin.	

**Table 2 diseases-13-00124-t002:** Baseline characteristics of the included patients.

Study ID	Age (Year), N (%)		Male Population, N (%)		Type of Area, N (%)	Type of Animal, N (%)	Bitten Organ, N (%)		Regimen of Vaccine, N (%)		Type of Injury, N (%)
Khoubfekr et al., 2024 [22]	The age ranged from 7 to 12 years old		2 (100)		NA	Dog 2 (100)	Upper limb 2 (100) Head injury 1 (50)		>24 h: 1 (50) <24 h: 1 (50)		NA
Davarani et al., 2023 [28]	Group 1 (<3) 99 (10.6) Group 2 (4–6) 225 (24.1) Group 3 (7–12) 609 (65.3)		620 (66.5)		Urban 835 (89.5) Rural 98 (10.5)	Dog 632 (67.7) Cat 237 (25.4) Hamster 44 (4.7) Other 20 (2.1)	Hand (including fingertips to wrist) 394 (42.2) Lower limbs (including legs/buttocks) 295 (31.6) Hands (including forearm, arm, shoulder) 84 (9) Head, face, and neck 86 (9.2) Chest, abdomen, and back 25 (2.7) Two organs of the body 39 (4.2) Three organs and more 10 (1.1)		<48 h: 868 (93) 48–72 h: 52 (5.6) 4–10 days: 12 (1.3) 11–20 days: 1 (0.1)		Puncture and scratching 573 (61.4) Scratch 345 (36.9) Perforation 8 (0.8) Crushing a part of the body 6 (0.6) Bone fracture 1 (0.10)
Khazaei et al., 2023 [18]	Group 1 (<5) 11,231 (4.31) Group 2 (5–15) 45,377 (17.42) Group 3 (16–30) 67,193 (25.80) Group 4 (31–45) 72,849 (27.97) Group 5 (46–60) 39,655 (15.22) Group 6 (+60) 24,165 (9.28)		201,164 (77.23)		Urban 146,868 (56.39) Rural 90,140 (34.61) Mobile 23,462 (9.01)	Domestic 256,138 (98.34) Wild 1614 (0.62) Redundant 2718 (1.04)	NA		<48 h: 253,185 (97.20) >48 h: 7285 (2.80)		Bone fracture 66 (0.03) Perforation 49,133 (18.89) Rupture 6634 (2.55) Segregation of the tissue 6259 (2.40)
Yıldırım et al., 2022 [30]	The mean and standard deviation 28.12 ± 21.60		466 (62.3)		NA	Dog 463 (61.9) Cat 259 (31.6) Wild animal 15 (2) Bat 2 (0.3) Other 9 (4.2)	NA		Administered vaccination: 477 (63.8) Not administered vaccination: 271 (36.2)		NA
Bay et al., 2021 [15]	The highest number was related to the age groups of 1 to 19 years, occupying 4629 (38)		9562 (78.5)		NA	NA	NA		No need for vaccines: 32 (0.3) Incomplete vaccine: 10,591 (86.9) Full vaccine (4 times): 1558 (12.8)		NA
Celiloglu et al., 2021 [38]			565 (62.4%)		NA	Cat 478 (52.8) Dog 413 (45.6) Bat 4 (0.4) Other species 11 (1.2)	NA		<24 h: 720 (79.4)		Scratch 428 (47.2) Bite 388 (42.8) Open wound contact 11 (1.2) Other types of trauma (e.g., abrasions) 79 (8.7)
Oztoprak et al., 2021 [43]	Group 1 (0–15) 730 (29.04) Group 2 (≥15) 1783 (70.95)		1599 (63.6)		Urban 2304 (91.68) Rural 209 (8.3)	Dogs 1539 (61.2) Cats 877 (34.9) Cattle 59 (2.3) Sheep/goat 13 (0.5) Bat 15 (0.6) Foxes 2 (0.1)	Upper extremity 1447 (57.58) Lower extremity 997 (39.67) Head and neck 112 (4.5) Abdomen 33 (1.3) Thorax 16 (0.6) Genital area 2 (0.08)		Five doses: 1017 (40.5) Four doses: 626 (24.9) 2-1-1 schema of vaccination: 372 (14.8)		Bite 1867 (74.3) Scratch 549 (21.8) Contact to open wound 21 (0.8) Other exposures 76 (3)
Porsuk et al., 2021 [31]	Group 1 (≤6) 458 (13.55) Group 2 (7–14) 563 (16.66) Group 3 (15–64) 2096 (62.04) Group 4 (≥65) 261 (7.72)		2097 (62.1)		Urban 2713 (80.3) Suburban 236 (7.0) Rural 429 (12.7)	Dogs 1865 (55.2) Cats 1483 (43.9) Mice 10 (0.3) Horses 3 (0.1) Bats 3 (0.1) Monkey and seagull 2 (0.05) Human 12 (0.4)	NA		One dose: 256 (7.6) Two doses: 215 (6.4) Three doses: 1129 (33.4) Four doses: 653 (19.3) Five doses: 1125 (33.3)		NA
Amiri et al., 2020 [32]	Group 1 (<7) 178 (4.33) Group 2 (7–18) 570 (13.8) Group 3 (19–30) 1564 (38.11) Group 4 (31–45) 1077 (26.24) Group 5 (46–60) 518 (12.6) Group 6 (>61) 197 (4.8)		3648 (88.89)		NA	Domestic 4081 (99.4) Wild and feral 23 (0.6) Dogs 2909 (70.88) Cats 997 (24.29) Others 198 (4.82)	Head and Neck 6 (0.1) Chest and abdomen 11 (0.3) Shoulder and hand 166 (4) Hips and buttocks 19 (0.5) Thighs and legs 211 (5.1) No 2997 (73)		Vaccinated: 1841 (44.9) Not vaccinated: 2263 (55.1)		NA
Can et al., 2020 [44]	Group 1 (0–18) 232 (33.5) Group 2 (19–36) 261 (37.6) Group 3 (37–54) 113 (16.3) Group 4 (55–72) 85 (12.3)		547 (79.2)		Urban 506 (73.2) Rural 185 (26.8)	Dogs 483 (69.9) Cats 171 (24.7) Horses 14 (2) Foxes 11 (1.6) Cattle 10 (1.4) Sheep 2 (0.3) Cow 10 (1.4)	Head 23 (3.3) Arm 385 (55.7) Body 90 (13.0) Leg 193 (27.9)		Complete vaccination: 227 (32.9) Incomplete vaccination: 464 (67.14)		Bite 510 (73.8) Scratch 159 (23.0) Contact 22 (3.2)
Janatolma Kan et al., 2020 [33]	Group 1 (1–9) 599 (11.0) Group 2 (10–19) 772 (14.0) Group 3 (20–29) 1347 (24.0) Group 4 (30–39) 1050 (19.0) Group 5 (40–49) 706 (13.0) Group 6 (50–59) 576 (10.2) Group 7 (>60) 568 (10.1)		4277 (76.3)		Urban 4239 (76.0) Rural 1365 (24.0)	Dog 4032 (72.0) Cats 1194 (21.3) Livestock 41 (0.7) Others 335 (6.0)	Upper limbs 2776 (49.5) Lower limbs 2666 (47.6) Both limbs 162 (3.0)		Three doses: 4594 (82.0) Five doses: 1009 (18.0)		NA
Rahmanian et al., 2020 [22]	Group 1 (<4) 8 (2.10) Group 2 (5–9) 27 (7.20) Group 3 (10–19) 74 (19.70) Group 4 (20–29) 84 (22.40) Group 5 (30–39) 86 (22.90) Group 6 (40–49) 96 (25.60)		314 (83.70)		Urban 203 (54.10) Rural 172 (45.90)	Dog 255 (68.00) Cat 101 (26.90) Hours 5 (1.30) Monkey (pet) 2 (0.50) Sheep 1 (0.30) Donkey 1 (0.30) Hamster 3 (0.80) Wild pigs 3 (0.80) Fox 2 (0.50) Wolf 1 (0.30) Hedgehog 1 (0.30)	Hand 103 (27.4) Leg 97 (25.8) Arm and forearm 85 (22.6) Ankle 56 (14.9) Shoulder 17 (4.53) Trunk 10 (2.66) Head/face 7 (1.86)		Three doses of vaccine: 268 (71.4) Five doses of vaccine: 107 (28.5) One dose of RIG: 109 (29.06)		NA
Rasooli et al., 2020 [25]	Ages ranged from 10 to 67 years old		7 (100)		NA	NA	The most common sites of injury were the hands and face, affecting 5 (71.43) of the individuals		NA		NA
Sarbazi et al., 2020 [34]	Group 1 (<10) 382 (12.6) Group 2 (10–20) 339 (11.2) Group 3 (20–30) 706 (23.3) Group 4 (30–40) 599 (19.8) Group 5 (40–50) 409 (13.5) Group 6 (>50) 597 (19.7)		2438 (80.4)		Urban 2094 (69.1) Rural 938 (30.9)	Carnivorous (Dog, Jackal, Fox) 1793 (59.2) Cat 1092 (36.0) Other 146 (4.8)	Upper limb 2006 (66.2) Lower limb 1026 (33.8)		<48 h: 2773 (92.5) >48 h: 259 (8.5)		Puncture of the wound 266 (8.8) No Puncture of the wound 2766 (91.2)
Hamta et al., 2019 [48]	Group 1 (<10) 231 (9.56) Group 2 (10–20) 377 (15.6) Group 3 (20–30) 612 (25.35) Group 4 (30–40) 532 (22.04) Group 5 (40–50) 286 (11.84) Group 6 (>50) 376 (15.57)		2009 (83.22)		Rural 474 (19.6) Urban 1897 (78.58)	Cattle (Horse, Donkey, Cow, Sheep, Camel, Goat) 75 (3.10) Carnivorous (Dog, Jackal, Pig, Fox) 1187 (49.17) Cat 1100 (45.56) Other 52 (2.15)	NA		>48 h: 305 (12.6) <48 h: 2109 (87.36)		Puncture wounds 283 (11.7) Scratches 2212 (91.6) Crush injuries 39 (1.6)
Kassiri et al., 2018 [35]	Group 1 (0–4) 180 (7.2) Group 2 (5–10) 333 (13.4) Group 3 (11–20) 436 (17.5) Group 4 (21–30) 543 (21.8) Group 5 (31–40) 410 (16.4) Group 6 (41–50) 266 (10.7) Group 7 (51–60) 235 (9.4) Group 8 (>61) 90 (3.6)		1910 (76.6)		Urban 1620 (65.0) Rural 873 (35.0)	Dog 1954 (78.4) Cat 432 (17.3) Others 107 (4.3)	Hands 1036 (41.6) Feet 1168 (46.9) Head and neck 143 (5.7) Trunk 146 (5.8)		Incomplete vaccination: 1531 (61.4) Complete vaccination: 962 (38.6)		NA
Khazaei et al., 2018 [29]	Group 1 (<5) 76 (5.25) Group 2 (5–15) 314 (21.69) Group 3 (16–30) 402 (27.76) Group 4 (31–45) 302 (20.86) Group 5 (46–60) 231 (15.95) Group 6 (>60) 123 (8.49)		1067 (80.60)		Urban 472 (32.60) Rural 976 (67.40)	Dogs 1088 (75.13) Cats 316 (21.8) Donkeys 16 (1.10) Rats 8 (0.55) Wolves 7 (0.48) Sheep 5 (0.35) Others 8 (0.55)	NA		<48 h: 1175 (81.14) >48 h: 273 (18.85)		NA
Babazade H. et al., 2016 [36]	The mean age of men (20.52 ± 14.72) and women (22.20 ± 14.11)		1240 (71.9)		Urban 409 (23.7) Rural 1 234 (71.5) Unknown 81 (4.8)	Pet dogs 1642 (95.2) Stray dogs 44 (2.6) Cats 21 (1.2) wolves 5 (0.3) Foxes 2 (0.1) other animals 10 (0.6)	Lower extremity 1 337 (77.6) Shoulders and hands 280 (16.2) Chest 66 (3.8) Head and neck 41 (2.4)		Three doses: 1607 (93.2) Five doses: 117 (6.8)		NA
Ramezankhani et al., 2016 [21]	Mean age 26.8 years (SD, ±13.1 years)	Mean age: 27.4 years (SD, ±13.9 years)	554 (78.9)	616 (82.4)	NA	NA	Head and neck 4 (30.8) Chest and abdomen 9 (52.9) Back 12 (50.0) Upper extremity 246 (49.3) Lower extremity 396 (47.5) Multiple locations 35 (56.5)	Head and neck 9 (69.2) Chest and abdomen 8 (47.1) Back 12 (50.0) Upper extremity 253 (50.7) Lower extremity 438 (52.5) Multiple locations 27 (43.5)	(Five doses of vaccine + Ig) 702 (48.45)	(2 + 1 + 1) 747 (51.55)	NA
Amiri et al., 2015 [37]	Group 1 (0–9) 127 (7.7) Group 2 (10–19) 228 (13.9) Group 3 (20–29) 330 (20.1) Group 4 (30–39) 295 (18.0) Group 5 (40–49) 266 (16.2) Group 6 (50–59) 228 (13.9) Group 7 (60) 169 (10.3)		1328 (75)		Urban 645 (39.3) Rural 998 (60.7)	History of dog bite was mentioned by 151 victims (9.2%) and 107 of them (70.9%) received rabies vaccine in previous bite	Lower extremity 899 (50.76) Upper extremity 723 (40.8) Abdominal and trunk 116 (6.5) Head, face, and neck 23 (1.30) Genital 2 (0.1)		NA		NA
Riabi et al., 2015 [39]	Group 1 (1–10) 49 (8) Group 2 (11–20) 82 (13.3) Group 3 (21–30) 168 (27.3) Group 4 (31–40) 78 (14.1) Group 5 (41–50) 70 (11.4) Group 6 (>50) 160 (26)		485 (78.7)		Urban 367 (59.57) Rural 230 (37.33)	Domestic dog 411 (66.7) Domestic cat 66 (10.7) Stray cat 60 (9.7) Stray dog 33 (5.4) Other animals 145 (23.5)	Hand 302 (49) Foot 228 (37) Hands and feet 22 (3.6) Legs and buttocks 3 (0.5) Hips 23 (3.7) Arm 3 (0.5) Side 6 (1.0) Neck 2 (0.3) Back 5 (0.8) Face 10 (1.6) Hands and face 1 (0.2) Abdomen 3 (0.5) Testis 1 (0.2) Head 2 (0.3) Thigh 1 (0.2) Leg and waist 1 (0.2) Unknown 3 (0.5)		Uncompleted vaccination: 517 (83.9) Completed vaccination: 99 (16.1)		NA
Poorolajal et al., 2015 [40]	Group 1 (<10) 2050 (9.08) Group 2 (11–20) 4145 (18.37) Group 3 (21–30) 7206 (31.93) Group 4 (31–40) 3741 (16.58) Group 5 (41–50) 2624 (11.63) (Group 6 (51–60) 1689 (7.48 Group 7 (61–70) 783 (3.47)		19,216 (84.51)		Urban 14,921 (66.27) Rural 7596 (33.73)	Dog 18,157 (81.71) Cat 2964 (13.34) Wolf 39 (0.18) Fox 69 (0.31) Jackal 8 (0.04) Donkey 67 (0.30) Cow 30 (0.14) Others 886 (3.99)	Head and neck 448 (2.06) Legs 9916 (45.62) Hands 9424 (43.36) Trunk 511 (2.35) Hand and foot 854 (3.93) Multiple organs 581 (2.67)		Three doses: 17,971 (78.94) Five doses: 4795 (21.06)		NA
Farahtaj et al., 2014 [26]	The age ranged from 11 months to 80 years (median, 21.5 years)		12 (75)		Urban 5 (31.25) Rural 11 (68.75)	Fox 2 (12.5) Dog 11 (68.75) Wolf 3 (18.75)	Head, face, trunk 10 (62.5) Upper extremities 8 (50) Abdomen, back, buttock 3 (18.75)		One dose: 1 (6.25) Three doses: 2 (12.50) Four doses: 13 (81.25)		NA
Karbeyaz et al., 2014 [50]	Group 1 (0–18) 159 (48.5) Group 2 (19–34) 96 (29.3) Group 3 (≥35) 73 (22.2)		235 (71.6)		NA	Dog 328 (100)	Lower extremity 161 (49.1) Upper extremity 102 (31.1) Head/neck/face 44 (13.4) Chest/abdomen/back 53 (16.1)		Vaccinated against rabies: 42 (12.8) Vaccinated against tetanus: 38 (11.6) No vaccination: 16 (4.9) Unknown: 270 (82.3)		Ecchymosis/abrasions/bruising/break, such as scratch or puncture in the skin, 240 (73.2) Laceration 59 (18.0) Flap-style injury 23 (7.0) Vascular Injury 6 (1.8)
Seri et al., 2014 [20]	Group 1 (0–15) 532 (31.6) Group 2 (15–60) 1030 (61.1) Group 3 (>60) 123 (7.3)		1089 (64.6)		NA	Dog 1133 (67.2) Cat 531 (31.5) Wild animal 15 (0.9) Rodent/Other 3 (0.2)	Head/neck 103 (6.1) Upper extremity 896 (53.2) Lower extremity 531 (31.5) Body 119 (7.1) Multiple sites 36 (2.1)		(2 + 1 + 1): 1420 (84.3) (Five doses of vaccine + Ig): 265 (15.7)		Bite 1380 (81.9) Scratch 298 (17.7) Lick 1 (0.1) Other 6 (0.4)
Charkazi et al., 2013 [41]	The mean and standard deviation of age 25.0 ± 17.8 years. The highest cases of bites were related to the age range of 11–15 years (2320 cases) (17.7)		9479 (72.1)		Rural 11,038 (84) Urban 2104 (16)	Dogs 12,895 (98.8) Cows 210 (1.6) Cats 39 (0.3) Camels 13 (0.1) Horses 13 (0.1) Donkeys 13 (0.1)	Leg 9136 (69.6) Body 2416 (12) Hand 1938 (9.2) Head, face, and neck 279 (2.1) Several organs 75 (0.6) Unclear 4 (0.03)		Complete vaccination: 6463 (72) Incomplete vaccination: 6679 (28)		NA
Taghvaii et al., 2013 [17]	Group 1 (<5) 392 (8.2) Group 2 (5–10) 1029 (7. 3) Group 3 (10–20) 3331 (23.7) Group 4 (20–30) 3488 (24.8) Group 5 (30–40) 2147 (15.3) Group 6 (40–50) 1496 (10.7) Group 7 (50 and >50) 2154 (15.4)		11,482 (81.8)		Urban 8633 (61.5) Rural 5404 (38.5)	Dog 10,848 (77.3) Cat 2504 (17.8) Wolf 11 (1) Jackal 6 (0.04) Fox 22 (2)	Hands 6197 (44.1) Legs 6166 (43.9) Body 1197 (8.5) Head and face 427 (3) Neck 50 (0.3%)		Incomplete vaccination: 11,672 (83.1) Complete vaccination: 2365 (16.9)		NA
Ghannad et al., 2012 [42]	The majority of animal bites were reported among the 10–19-year-old (143.9 per 10,000 population) age group		3032 (68.3)		Urban 1017 (23) Rural 3227 (73)	Dog 3942 (89.2) Cats 221 (5) Wolves 31 (0.7) Jackals 27 (0.6) Foxes 13 (0.3) Other animals 186 (4.2)	Lower extremities 3165 (71.6) Hands 884 (20) Trunk 265 (6) Head and face 88 (2) Neck 18 (0.4)		Incomplete vaccination: 3596 (81.3) Complete vaccination: 824 (18.7)		NA
Ansari et al., 2011 [23]	Ages ranged from 22 to 25 years old		2 (100%)		NA	Dogs 2 (100)	Lower extremity 2 (100)		Five doses: 2 (100)		Bite 2 (100)
Bijari et al., 2011 [46]	Mean age of 33.4± 19.5 years (range from 1 to 90 years)		1300 (78.2)		Rural 595 (35.8) Urban 1067 (64.2)	Domestic 1435 (86.3%) Wild 73 (4.4) Stray 154 (9.3)	NA		Complete: 301 (18.1) Incomplete: 1361 (81.9)		NA
Najafi et al., 2009 [49]	Group 1 (0–4) 466 (2.01) Group 2 (5–9) 1822 (7.86) Group 3 (10–19) 6605 (28.50) Group 4 (20–29) 5064 (21.85) Group 5 (30–39) 3387 (14.61) Group 6 (40–49) 2816 (12.15) Group 7 (>50) 3008 (12.98)		18,445 (79.6)		Urban 11,797 (36.77) Rural 20,282 (63.22)	NA	Lower extremities 16,979 (52.93%) Upper extremities 12,671 (39.5) Trunk 1841 (5.74) Face 577(1.8) Neck 119 (0.37)		NA		Bite 32,077 (99.99) Scratch 2 (0.006)
Sheikholeslami et al., 2009 [51]	Children: 277 (18) Adults: 1265 (82)		1310 (85)		Urban 694 (45) Rural 848 (55)	Dogs 1141 (74) Cats 355 (23) Other animals 46 (3)	Hand 62 (4) Foot 524 (34) Head 77 (5) Trunk 62 (4) Mixed sites of injury 817 (53)		The mean time delay from injury to initial management for both sexes was 15.1 (SD 29.8) hours		NA
Kilic et al., 2006 [16]	The ages ranged from 0 to 85 years (median, 25 years), and 43.5% of them were younger than 20 years		1046 (66.7)		Urban 690 (44) Rural 879 (56)	Dogs 1087 (96.2) Cats 298 (19) Other (cattle, rat, etc.) 112 (79.5)	Upper extremities 493 (68.3) Lower extremities 457 (68.7) Other 232 (71.1)		Vaccinated: 1067 (68) Not vaccinated: 502 (32)		NA
Sengoz et al., 2006 [47]	One-third of the cases (2422) were under 15 years of age		5450 (75)		NA	Dog bite 5390 (74) Cat bite 1850 (25) Other animals 25 (0.3)	Head/neck wounds 298 (4) Trunk/extremity wounds 4610 (63) Hand wounds 2360 (32)		Complete vaccination: 1750 (24) Incomplete vaccination: 5516 (76)		NA
Bizri et al., 2000 [27]	The ages ranged from 11 to 77 years.		5 (62.5)		NA	Dog 5 (62.5) Other 3 (37.5)	NA		Vaccinated: 1 (12.5) Not vaccinated: 7 (87.5)		NA
Tabbara et al., 1995 [24]	The age ranged from 1.5 to 7 years old		0 (0)		Desert 2 (100)	Fox 2 (100)	Face and eyelid 2 (100)		>48 h: 2 (100)		NA

**Table 3 diseases-13-00124-t003:** Summary of the baseline characteristics of the included patients.

Category	Most Common Findings	Range/Notes
Age Groups Affected	Highest exposure: young adults (20–39 years) consistently showed the highest rabies exposure rates, likely due to occupational or outdoor activities Children (0–12 years): significant exposure, especially in regions with stray dog populations Elderly (60+): lower but non-negligible exposure, possibly due to reduced mobility or rural residency	Many studies used age groupings; children <15 highly represented in several cases
Sex (Male %)	Mean ~70–80% male	Male predominance observed in nearly all studies
Type of Area	Urban (dominant in most studies)	Mixed urban/rural settings also reported
Main Animal Involved	Dogs (most common), followed by cats	Other animals: foxes, bats, monkeys, livestock
Most Bitten Body Parts	Lower extremities, upper limbs, hands	Some studies also reported multiple sites or head/neck injuries
Time to Vaccination	Majority < 48 h	Some delays >48 h in rural or underserved areas
Vaccination Regimen	3- or 5-dose regimens most frequent	Some also reported incomplete vaccination and RIG administration
Injury Type	Bites (most common), scratches, punctures	Severe injuries (fractures, crush injuries) reported in a few studies

**Table 4 diseases-13-00124-t004:** Safety of the included vaccines.

Study ID	Seri et al., 2014 [20]		Ramezankhani et al., 2016 [21]	
Vaccine	Verorab	Abhayrab	PVRV	PCECV
Local pain, N (%)	23 (3.02)	87 (9.4)	27 (3.9)	28 (3.8)
Local redness and swelling, N (%)	3 (0.39)	19 (2.06)	9 (1.3)	8 (1.1)
Local numbness, N (%)	0 (0)	8 (0.86)	NA	
Itching at site of admin., N (%)	3 (0.39)	9 (0.97)	7 (1)	1 (0.1)
Swelling, N (%)	NA		4 (0.6)	2 (0.3)
Bruising, N (%)	NA		0 (0)	4 (0.5)
Headache, N (%)	27 (3.55)	95 (10.28)	8 (1.4)	16 (2.5)
Dizziness, N (%)	6 (0.78)	42 (4.54)	5 (0.8)	6 (0.9)
Fever, N (%)	38 (4.99)	229 (24.78)	11 (1.9)	10 (1.6)
Itching, N (%)	5 (0.65)	13 (1.4)	0 (0)	1 (0.2)
Weakness, N (%)	38 (4.99)	140 (15.15)	5 (0.8)	11 (1.7)
Abdominal pain, N (%)	13 (1.71)	30 (3.24)	4 (0.7)	3 (0.5)
Lymphadenopathy, N (%)	0 (0)	3 (0.324)	NA	
Nausea, N (%)	19 (2.49)	52 (5.62)	4 (0.7)	7 (1.1)
Vomiting, N (%)	2 (0.26)	16 (1.73)	4 (0.7)	7 (1.1)
Myalgia, N (%)	7 (0.91)	57 (6.17)	6 (1.0)	5 (0.8)
Cough, N (%)	9 (1.18)	11 (1.19)	NA	
Insomnia, N (%)	0 (0)	19 (2.05)	NA	
Numbness, N (%)	0 (0)	12 (1.29)	NA	
Irregular menstruation, N (%)	0 (0)	3 (0.324)	NA	
Decreased libido, N (%)	0 (0)	1 (0.11)	NA	
Rash, N (%)	2 (0.26)	9 (0.97)	NA	
Sore throat, N (%)	8 (1.05)	19 (2.05)	NA	
Nasal discharge, N (%)	8 (1.05)	2 (2.27)	NA	
Arthralgia, N (%)	14 (1.83)	69 (7.47)	6 (1)	3 (0.5)
Sweating, N (%)	NA		3 (0.5)	4 (0.6)
Hypotension, N (%)	NA		0 (0)	1 (0.2)

**Table 5 diseases-13-00124-t005:** Details of the quality assessment of the included studies.

Study ID	Clearly Stated Aim	Consecutive Patients	Prospective Collection of Data	Endpoints	Assessment of Endpoint	Follow-Up Period	Loss < 5%	Study Size	Adequate Control Group	Contemporary Group	Baseline Control	Statistical Analyses	MINORS Score
Khoubfekr et al., 2024 [22]	2	1	2	2	1	2	2	1	0	0	0	0	13
Davarani et al., 2023 [28]	2	2	1	2	2	2	2	2	0	0	0	0	15
Khazaei et al., 2023 [18]	2	2	1	2	2	2	2	2	0	0	0	0	15
Yıldırım et al., 2022 [30]	2	2	1	2	2	2	2	2	0	0	0	0	15
Bay et al., 2021 [15]	2	2	1	2	2	2	2	2	0	0	0	0	15
Celiloglu et al., 2021 [38]	2	2	1	2	2	2	2	2	0	0	0	0	15
Oztoprak et al., 2021 [43]	2	2	1	2	2	2	2	2	0	0	0	0	15
Porsuk et al., 2021 [31]	2	2	1	2	2	2	2	2	0	0	0	0	15
Amiri et al., 2020 [32]	2	2	1	2	2	2	2	2	0	0	0	0	15
Can et al., 2020 [44]	2	2	1	2	2	1	2	2	0	0	0	0	15
Janatolma Kan et al., (2020) [33]	2	2	1	2	2	2	2	2	0	0	0	0	15
Rahmanian et al., 2020 [22]	2	2	1	2	2	2	2	2	0	0	0	0	15
Rasooli et al., 2020 [25]	2	1	1	2	1	2	2	1	0	0	0	0	12
Sarbazi et al., 2020 [34]	2	2	1	2	2	2	2	2	0	0	0	0	15
Hamta et al., 2019 [48]	2	2	1	2	2	2	2	2	0	0	0	0	15
Kassiri et al., 2018 [35]	2	2	1	2	2	1	2	2	0	0	0	0	14
Khazaei et al., 2018 [29]	2	2	1	2	2	2	2	2	0	0	0	0	15
BabazadeH. et al., 2016 [36]	2	2	1	2	2	2	2	2	2	2	2	2	23
Ramezankhani et al., 2016 [21]	2	2	2	2	2	2	2	2	2	2	2	2	24
Amiri et al., 2015 [37]	2	2	1	2	2	1	2	2	0	0	0	0	14
Riabi et al., 2015 [39]	2	2	1	2	2	1	2	2	0	0	0	0	14
Poorolajal et al., 2015 [40]	2	2	1	2	2	2	2	2	0	0	0	0	15
Farahtaj et al., 2014 [26]	2	1	1	2	1	2	2	1	0	0	0	0	12
Karbeyaz et al., 2014 [50]	2	2	1	2	2	1	2	2	0	0	0	0	14
Seri et al., 2014 [20]	2	2	2	2	2	1	2	2	2	2	2	2	23
Charkazi et al., 2013 [41]	2	2	1	2	2	1	2	2	0	0	0	0	14
Taghvaii et al., 2013 [17]	2	2	1	2	2	2	2	2	0	0	0	0	15
Ghannad et al., 2012 [42]	2	2	1	2	2	1	2	2	0	0	0	0	14
Ansari et al., 2011 [23]	2	1	2	2	1	2	2	1	0	0	0	0	13
Bijari et al., 2011 [46]	2	2	1	2	2	2		2	0	0	0	0	15
Najafi et al., 2009 [49]	2	2	1	2	2	2		2	0	0	0	0	15
Sheikholeslami et al., 2009 [51]	2	2	1	2	2	2		2	0	0	0	0	15
Kilic et al., 2006 [16]	2	2	1	2	2	2		2	0	0	0	0	15
Sengoz et al., 2006 [47]	2	2	1	2	2	2		2	0	0	0	0	15
Bizri et al., 2000 [27]	2	1	1	2	1	2		1	0	0	0	0	12
Tabbara et al., 1995 [24]	2	1	2	2	1	2		1	0	0	0	0	13

## Data Availability

No new data were created or analyzed in this study. Data sharing does not apply to this article.

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
