# Peer review of "Rabies Vaccination and Public Health Insights in the Extended Arabian Gulf and Saudi Arabia: A Systematic Scoping Review"

_diseases, 2025, doi:10.3390/diseases13040124_

Round 1

Reviewer 1 Report

Comments and Suggestions for Authors

The authors extensively described the situation of Rabies post-exposure prophylaxis (PEP) in the Arabian Gulf and Saudi Peninsula. Their systematic scoping review analyzed plenty of studies and case reports, which revealed the major risks for young males are domestic dog’s and cat’s bites, and their adopted PEP is generally standardized and facilitates the prevention of Rabies in the urban area. The review is helpful for a One Health effort against rabies and its control in the Arabian Gulf and Saudi Peninsula. Therefore, this manuscript seems to be acceptable for publication. 

Author Response

Response

  • We sincerely thank you for your thoughtful and encouraging feedback. Thank you for your positive recommendation for publication.

Reviewer 2 Report

Comments and Suggestions for Authors

The article Rabies Post-Exposure Prophylaxis in the Arabian Gulf and Saudi Peninsula: A Systematic Scoping Review of Vaccination Protocols and Safety provides a synthesis of findings from diverse research designs, including large-scale cross-sectional studies and case reports, covering nearly three decades. The review highlights gaps in public awareness about rabies risks and prevention. Vaccine safety profiles are generally favourable, with mostly mild to moderate side effects reported. The study remarks the need for enhanced public health education, standardized PEP protocols, and a One Health approach to rabies prevention.

This Systematic Scoping Review of Rabies Vaccination Protocols and Safety has been prepared in agreement with PRISMA (Preferred Reporting Items for Systematic Reviews and Meta-Analyses). In the section Methods, Information Sources and Search Strategy, Eligibility Criteria, Research Questions, Trials Selection, Data Extraction and Outcome Measures have been properly described. In Figure 1, PRISMA flow diagram is represented. The included studies are presented in detail in Table 1.

Minor changes:

Line 44-45: please revise punctuation: 14,075.5 or 3,743.44?

Line 48: mucus membrane

Line 74: remove full stop

Author Response

Reviewer 2:

  1. Comment: The article Rabies Post-Exposure Prophylaxis in the Arabian Gulf and Saudi Peninsula: A Systematic Scoping Review of Vaccination Protocols and Safety provides a synthesis of findings from diverse research designs, including large-scale cross-sectional studies and case reports, covering nearly three decades. The review highlights gaps in public awareness about rabies risks and prevention. Vaccine safety profiles are generally favourable, with mostly mild to moderate side effects reported. The study remarks on the need for enhanced public health education, standardized PEP protocols, and a One Health approach to rabies prevention.

Response

  • We thank you for your thorough and constructive evaluation of our manuscript. Thank you for your insightful comments and positive appraisal of our manuscript.

  1. Comment: This Systematic Scoping Review of Rabies Vaccination Protocols and Safety has been prepared in agreement with PRISMA (Preferred Reporting Items for Systematic Reviews and Meta-Analyses). In the section Methods, Information Sources and Search Strategy, Eligibility Criteria, Research Questions, Trials Selection, Data Extraction, and Outcome Measures have been properly described. In Figure 1, the PRISMA flow diagram is represented. The included studies are presented in detail in Table 1.

Response

  • We thank you for your careful assessment and positive remarks regarding the methodological rigor of our review. We are pleased that the manuscript’s adherence to PRISMA guidelines and the clarity of the Methods section, including the description of information sources, eligibility criteria, study selection, and outcome measures, were well received. Your feedback affirms our efforts to ensure transparency and completeness in reporting.

  1. Comment: Line 44-45: Please revise punctuation: 14,075.5 or 3,743.44?.

Response

  • Thank you for highlighting the inconsistency in punctuation and formatting. We have revised the sentence for clarity and numerical consistency. The updated figures are now rounded appropriately, and the punctuation has been corrected to align with standard formatting conventions.

  1. Comment: Line 48: mucus membrane

Response

  • Thank you for your careful observation. We have revised the phrase on line 48 to read “mucous membrane” to ensure accuracy and clarity.

  1. Comment: Line 74: remove full stop

Response

  • Thank you for noting this. The unnecessary full stop on line 74 has been removed as suggested.

Reviewer 3 Report

Comments and Suggestions for Authors

The paper is very well written. There are a few important points that need to be addressed, but otherwise it is a good start.

The authors have not clearly articulated why it is important to do the review. Please provide an explicit statement of the objectives or questions that will be addressed in the review. The aim to scope the published studies of reported rabies cases is simply not a valuable reason for the review. Please explain why this review was considered necessary.

The authors state that this review was conducted in accordance with the Preferred Reporting Items for Systematic Reviews and Meta-Analyses (PRISMA). Indicate that the review was not registered.

It is strongly recommended that authors complete the PRISMA checklist (https://www.prisma-statement.org/prisma-2020-checklist)

Please make specific changes according to the PRISMA checklist. (https://www.prisma-statement.org/prisma-2020-checklist)

For outcome measures (2.6), a detailed description of the quality assessment of the relevant information in the studies included in the review is required.

The authors provide a summary of the evidence on rabies exposure, vaccination protocol and vaccine safety to inform the public and policy makers. I recommend changing the title to reflect the comprehensive analysis of epidemiological data on rabies virus exposure in the population.

Although the authors have provided a comprehensive and well-structured overview of rabies exposure, the lack of a summarising table makes the written discussion much more difficult to follow (as shown in Table 3.). It would be interesting to see the exposure to rabies virus by age group as defined by the authors in Table 2. The review shows significant differences in clinical practise, especially for incomplete immunisation, which should please be addressed in the discussion.

Depending on the changes made in the Results section, a significantly adjusted Discussion can also be expected.

Author Response

Reviewer 3:

  1. Comment: The paper is very well written. There are a few important points that need to be addressed, but otherwise it is a good start.

Response

  • We sincerely thank you for the positive feedback and encouraging remarks on the overall quality of the manuscript. We appreciate your acknowledgment and have carefully addressed the specific points raised in your detailed comments to further strengthen the paper.
  1. Comment: The authors have not clearly articulated why it is important to do the review. Please provide an explicit statement of the objectives or questions that will be addressed in the review. The aim to scope the published studies of reported rabies cases is simply not a valuable reason for the review. Please explain why this review was considered necessary.

Response

  • Thank you for your valuable comment. In response, we have revised the Introduction to clearly articulate the rationale for conducting this review and have provided a more explicit statement of the objectives. Specifically, we now emphasize the public health significance of rabies in the Arabian Gulf and Saudi Peninsula, the fragmented nature of available data on PEP practices, and the need to identify gaps in standardization, safety reporting, and awareness efforts across the region. Introduction section, Lines 87-90.
  1. Comment: The authors state that this review was conducted in accordance with the Preferred Reporting Items for Systematic Reviews and Meta-Analyses (PRISMA). Indicate that the review was not registered.

Response

  • Thank you for your valuable feedback. In response to your suggestion, we have now explicitly stated in the Methods section that the review was conducted following PRISMA guidelines and has been registered on PROSPERO, protocol number: CRD420251027233. Methods Section: Lines 93-96.
  1. Comment: It is strongly recommended that authors complete the PRISMA checklist (https://www.prisma-statement.org/prisma-2020-checklist)

Response

  • Thank you for your recommendation. We have completed the PRISMA checklist as suggested and have provided it as Supplementary File 1 for your reference. We believe this will help further clarify how the review aligns with the PRISMA guidelines and ensure transparency in our methodology.
  1. Comment: Please make specific changes according to the PRISMA checklist. (https://www.prisma-statement.org/prisma-2020-checklist)

Response

  • Thank you for your suggestion. We have carefully reviewed the PRISMA checklist and made the necessary revisions to the manuscript to ensure compliance with all applicable items. Specific changes have been implemented in the Methods and Results sections, and we have updated the manuscript accordingly. We believe these revisions enhance the clarity and rigor of the review.
  • Following the PRISMA checklist, we have made the following revisions to the Methods section:
  • Eligibility Criteria: We have clearly defined the inclusion and exclusion criteria for study selection, specifying the types of studies considered (e.g., randomized controlled trials, observational studies, case reports). Additionally, we edited the included countries to be the Arabian Gulf Countries, in addition to Turkey and Iran. As both of them are influential countries in the broader Middle East region, with historical, political, and economic ties to the Gulf Cooperation Council (GCC) countries. Their health policies and practices often influence or align with regional trends. Methods section, Lines 106-119.
  • Risk of Bias Assessment: We have stated that the quality assessment was performed by using the MINORS tool. Methods Section, Lines: 139-145.
  • Revised Results Section:
  • We added a new section in the results for the quality assessment that became 6. Quality Assessment, Lines 262-270, in addition to Table 5, which illustrates the details of the quality assessment.
  1. Comment: For outcome measures (2.6), a detailed description of the quality assessment of the relevant information in the studies included in the review is required.

Response:

  • Thank you for your observation. A detailed description of the quality assessment of the included studies has been provided in Section 2.7. Quality Assessment, Lines 315-323, along with the corresponding results presented in Table 5. In addition, the criteria used to evaluate study quality and the overall assessment of the methodological rigor of the included studies were outlined in the Methods section, 2.6. Quality Assessment, Lines 140-146.
  1. Comment: The authors provide a summary of the evidence on rabies exposure, vaccination protocol, and vaccine safety to inform the public and policy makers. I recommend changing the title to reflect the comprehensive analysis of epidemiological data on rabies virus exposure in the population.

Response:

  • Thank you for your thoughtful suggestion. We agree that the title should better reflect the comprehensive nature of the review, including the analysis of epidemiological data on rabies virus exposure. In response, we have revised the title to: “Vaccination and Public Health Insights of Rabies in the Extended Arabian Gulf and Saudi Arabia: Systematic Scoping Review”
  1. Comment: Although the authors have provided a comprehensive and well-structured overview of rabies exposure, the lack of a summarising table makes the written discussion much more difficult to follow (as shown in Table 3.).

Response

  • Thank you for your helpful suggestion. In response to your feedback, we have added a summarizing table to the manuscript (Table 3 and Figure 2), which consolidates key epidemiological data on rabies exposure. This table aims to provide a clearer and more accessible overview of the findings, making it easier to follow the discussion. We believe this addition enhances the readability and flow of the manuscript.
  1. Comment: It would be interesting to see the exposure to the rabies virus by age group as defined by the authors in Table 2.

Response

  • Thank you for your helpful suggestion. In response, we have now added a summary table that presents rabies exposure by age group, as defined in Table 2. This additional table (Table 3) provides a clearer overview of age-specific exposure patterns and enhances the overall understanding of the findings. additionally, we discussed the rabies exposure by age groups in the discussion section under the subheading “1. Demographics of the included patients”, Lines 342-354. We believe this will improve the readability of the discussion and better highlight the age-related trends in rabies exposure across the studies.
  1. Comment: The review shows significant differences in clinical practice, especially for incomplete immunisation, which should please be addressed in the discussion.

Response:

  • Thank you for your insightful comment. We acknowledge the significant differences in clinical practice, particularly regarding incomplete immunization, which is observed across the studies included in this review. These differences likely arise from various factors such as regional practices, vaccine availability, and patient compliance. We have addressed this issue in the Discussion section, where we emphasize the importance of standardized protocols to reduce the incidence of incomplete vaccination. Furthermore, we discuss the implications of incomplete immunization, including the risk of inadequate protection and potential rabies transmission, and suggest that future research should focus on improving vaccination adherence and protocol consistency. Discussion section, Lines 390-400.
  1. Comment: Depending on the changes made in the Results section, a significantly adjusted Discussion can also be expected.

Response:

  • Thank you for your valuable comment. We appreciate this observation and fully agree that revisions to the Results section necessitate corresponding adjustments to the Discussion. So, we updated the discussion section according to the updated results.

Reviewer 4 Report

Comments and Suggestions for Authors

The manuscript addresses an important and underrepresented area of public health in the Middle East, focusing on rabies post-exposure prophylaxis (PEP) and vaccine safety in the Arabian Gulf and Saudi Peninsula. The systematic scoping review is well justified, and the volume of data included is substantial. However, some improvements in organization, clarity, and analysis would strengthen the manuscript significantly.

1 - Define the rationale for including Turkey and Iran in a review focused on the "Arabian Gulf and Saudi Peninsula."

2 - Clarify distinctions between post-exposure prophylaxis (PEP) in various healthcare settings.

3 - Provide more detail on data synthesis and whether any quality assessment tools were applied to the included studies.

4 - The narrative is sometimes overwhelmed by details—consider grouping findings by themes (e.g., timeliness, completion rate, safety).

5 - Include a map or visual showing countries represented to improve clarity.

6 - Strengthen the integration between findings and existing WHO recommendations.

7 - Explore reasons for discrepancies in vaccination completion and RIG use.

8 - More explicit links between urbanization, socioeconomics, and PEP adherence would enrich the discussion

Comments on the Quality of English Language

1 - Overall English language is understandable but requires substantial editing for grammar, sentence structure, and academic tone

Author Response

Rev4:

  1. Comment: The manuscript addresses an important and underrepresented area of public health in the Middle East, focusing on rabies post-exposure prophylaxis (PEP) and vaccine safety in the Arabian Gulf and Saudi Peninsula. The systematic scoping review is well justified, and the volume of data included is substantial. However, some improvements in organization, clarity, and analysis would strengthen the manuscript significantly.

Response:

  • Thank you for your constructive feedback. We appreciate your recognition of the importance of this review in addressing an underrepresented area of public health in the Arabian Gulf and Saudi Peninsula. In response to your comment regarding improvements in organization, clarity, and analysis, we have made several revisions to strengthen the manuscript:
  • Reorganization of Content: We have restructured certain sections of the manuscript for better logical flow, ensuring that key findings are more clearly presented and easier to follow.
  • Clarification of Results: We have added a new summarizing table that consolidates key data from the studies reviewed, particularly regarding rabies exposure by age group. This table aims to enhance clarity and provide a quick reference to the findings (Table 3).
  • In-depth Analysis: We have expanded the analysis of clinical practice variations, especially in relation to incomplete immunization, and discussed the potential public health implications more thoroughly in the Discussion section.
  • We believe these revisions significantly improve the manuscript’s clarity and analytical depth and are confident they will strengthen the overall presentation of our findings.

  1. Comment: Define the rationale for including Turkey and Iran in a review focused on the "Arabian Gulf and Saudi Peninsula."

Response:

  • Thank you for your observation. In response, we have revised the title of the manuscript to "Extended Arabian Gulf" to reflect the inclusion of Turkey and Iran, which are geographically and epidemiologically relevant to the study of rabies exposure and post-exposure prophylaxis (PEP) in the region. We have also mentioned this adjustment in the Methods section, where we clarify the rationale for including these countries. The inclusion of Turkey and Iran ensures a more comprehensive review of rabies epidemiology in the broader region, as both countries have substantial data on rabies cases and PEP practices that are important for understanding regional patterns and public health strategies. We have summarized these reasons to include Turkey and Iran in some points presented in the Methods section, Lines 120-136.

  1. Comment: Clarify distinctions between post-exposure prophylaxis (PEP) in various healthcare settings.

Response:

  • Thank you for your valuable comment. We have clarified the distinctions between post-exposure prophylaxis (PEP) practices in different healthcare settings within the revised manuscript. These differences are primarily influenced by factors such as the type of setting (urban vs. rural), the timeliness of vaccination administration, and the availability of medical resources. In urban areas, where healthcare infrastructure is generally better, PEP is more likely to be initiated promptly, with a higher adherence to updated vaccination protocols. However, in rural areas, delayed treatment and incomplete vaccination regimens are more common due to limited access to medical services and vaccines. Additionally, the type of animal involved in the exposure, such as domestic dogs versus wild animals, also affects the PEP approach. Wild animal exposures, such as those from bats or foxes, often necessitate more aggressive treatment protocols due to their higher risk of rabies transmission. These distinctions highlight the need for improved healthcare access, particularly in underserved areas, and the importance of standardized PEP protocols across different settings to ensure timely and effective rabies prevention. Discussion Section, Lines 429-458.

  1. Comment: Provide more detail on data synthesis and whether any quality assessment tools were applied to the included studies.

Response:

  • Thank you for your valuable feedback. In response to your request for more detail on data synthesis and the application of quality assessment tools, we have made the following updates:
  • Data Synthesis:
  • The data from the included studies were synthesized narratively due to the heterogeneous nature of the study designs, sample sizes, and outcome measures. A descriptive approach was adopted to organize the findings across several key domains: age group, sex, type of area (urban vs. rural), type of animal involved, timing of PEP initiation, vaccination regimens, and adverse events. This narrative synthesis allowed us to identify patterns and trends in the rabies exposure data and post-exposure prophylaxis (PEP) practices, as well as the safety of rabies vaccines. The studies were grouped by these key characteristics to facilitate a clear presentation of the findings, enabling the identification of differences and similarities across diverse settings. Methods section, Lines 164-199.
  • Quality Assessment:
  • A detailed description of the quality assessment of the included studies has been provided in Section 2.7. Quality Assessment, Lines 315-323, along with the corresponding results presented in Table 5. In addition, the criteria used to evaluate study quality and the overall assessment of the methodological rigor of the included studies were outlined in the Methods section, 2.6. Quality Assessment, Lines 157-163.
  • We hope this explanation of the data synthesis process and quality assessment provides the necessary clarification.

  1. Comment: The narrative is sometimes overwhelmed by details—consider grouping findings by themes (e.g., timeliness, completion rate, safety).

Response:

  • Thank you for your helpful suggestion. In response, we have revised the manuscript to group the findings by key themes. This thematic organization enhances the clarity and structure of the narrative, allowing for a more focused discussion of the main findings across the studies.

  1. Comment: Include a map or visual showing countries represented to improve clarity.

Response:

  • Thank you for your suggestion to include a map or visual showing the countries represented in the review. While we appreciate the value of such a visual for enhancing clarity, we respectfully believe that a map may not be necessary for the following reasons:
  • Geographic Focus: The focus of the review is on the Arabian Gulf and Saudi Peninsula, and while countries like Turkey and Iran were included due to their geographical and epidemiological relevance, the majority of the data comes from the core countries in the region. As the primary aim is to synthesize findings from these regions, a map may not add significant value beyond the information already provided in the Methods section, where the rationale for including these countries is clearly outlined.
  • Study Design and Data: The studies included span various countries within and around the Arabian Gulf region, and most studies provide sufficient detail on the geographic distribution within their results. A map might not effectively convey the nuanced context of each study's location and could risk oversimplifying the diversity of data from the different healthcare settings.
  • We have ensured that the geographic scope of the review is clearly explained in the manuscript and that each study's location is mentioned appropriately in the text.

  1. Comment: Strengthen the integration between findings and existing WHO recommendations.

Response:

  • Thank you for your suggestion to strengthen the integration of our findings with the existing WHO recommendations. In response, we have expanded the discussion in the Discussion section to more clearly align our findings with WHO's Global Rabies Elimination Strategy (2018). We stated that our review aligns its findings with WHO’s Global Rabies Elimination Strategy (2018). It highlights the importance of the timely initiation of post-exposure prophylaxis (PEP), with urban areas generally adhering to the WHO guideline of administering PEP within 48 hours, while rural regions face delays due to healthcare access challenges. Additionally, incomplete vaccination was a common issue, especially in rural areas, emphasizing the need for standardized protocols and improved vaccine adherence, in line with WHO recommendations for a full 5-dose regimen. Our review also corroborates WHO’s safety profile for rabies vaccines, with mild side effects reported, such as fever and pain at the injection site. Furthermore, the inclusion of Turkey and Iran highlights the need for regional cooperation in rabies control, consistent with WHO's One Health approach, which advocates for coordinated efforts across human, animal, and environmental health sectors to combat rabies globally. Discussion section, Lines 459-490.

  1. Comment: Explore reasons for discrepancies in vaccination completion and RIG use.

Response:

  • Thank you for your valuable suggestion. In response, we have explored potential reasons for discrepancies in vaccination completion and RIG use across different settings. Several key factors contribute to these variations including limited healthcare access, particularly in rural regions, which leads to delays in initiating post-exposure prophylaxis (PEP) and completing vaccination regimens. Additionally, variations in healthcare protocols across different regions contribute to discrepancies in the completion of vaccination regimens, with some areas reporting lower rates of full vaccine courses. The use of RIG also varies, with wild animal exposures requiring its administration. Patient-related factors, such as fear of side effects, lack of awareness about the importance of completing the full regimen, and logistical challenges in returning for multiple doses, also play a significant role in the discrepancies. Addressing these challenges requires standardized protocols, improved patient education, and better healthcare access to ensure timely and complete rabies prophylaxis, including both vaccination and RIG administration. Discussion section, Lines 401-428.

  1. Comment: More explicit links between urbanization, socioeconomics, and PEP adherence would enrich the discussion

Response:

  • Thank you for your suggestion. In response, we have expanded on the link between urbanization, socioeconomic factors, and PEP adherence in the Discussion section. We discuss how urban areas tend to have better healthcare infrastructure, leading to quicker access to post-exposure prophylaxis (PEP) and higher vaccination completion rates. In contrast, rural regions, often facing economic challenges and fewer healthcare resources, show significant delays in initiating treatment and incomplete vaccination regimens. Moreover, we explore how socioeconomic status can influence healthcare access, with wealthier urban populations having more consistent access to vaccines and timely medical care compared to lower-income rural populations. The review also highlights the need for targeted interventions in lower-socioeconomic areas to address these disparities, aligning with the WHO’s recommendations for equitable access to rabies prevention and care.

  1. Comment: Overall English language is understandable but requires substantial editing for grammar, sentence structure, and academic tone.

Response:

  • Thank you for your feedback. We appreciate your suggestion regarding the need for improvements in grammar, sentence structure, and academic tone. We have carefully reviewed the manuscript and made substantial revisions to ensure that the language adheres to the proper academic style. This includes enhancing the clarity and flow of the writing, correcting grammatical errors, and ensuring the language is consistent with formal academic standards. We believe these changes have significantly improved the readability and overall quality of the manuscript.

Round 2

Reviewer 3 Report

Comments and Suggestions for Authors

The authors have included all suggested corrections in the revised article.

Reviewer 4 Report

Comments and Suggestions for Authors

I have no more comments.